

# The risky middle of the road - probabilities of triggering climate tipping points and how they increase due to tipping points within the Earth's carbon cycle

Jakob Deutloff[1, 2, 3], Hermann Held[4], and Timothy M Lenton[1]

[1]Global Systems Institute, University of Exeter, Exeter, UK.
[2]Center for Earth System Research and Sustainability (CEN), Meteorological Institute, Universität Hamburg, Hamburg, Germany.
[3]International Max Planck Research School on Earth System Modelling (IMPRS-ESM), Hamburg, Germany.
[4]Research Unit Sustainability and Global Change and Center for Earth System Research and Sustainability, Universität Hamburg, Germany.

**Correspondence:** Jakob Deutloff (jakob.deutloff@mpimet.mpg.de)

**Abstract.** We investigate the probabilities of triggering climate tipping points under various shared socioeconomic pathways (SSPs), and how they are altered by including the additional carbon emissions that could arise from tipping points within the Earth's carbon cycle. Crossing of a climate tipping point at a threshold level of global mean surface temperature (threshold temperature), would commit the affected subsystem of the Earth to abrupt and largely irreversible changes with negative impacts on human well-being. However, it remains unclear which tipping points would be triggered under the different SSPs, due to uncertainties in the climate sensitivity to anthropogenic greenhouse gas emissions, the threshold temperatures of climate tipping points, and the response of tipping points within the Earth's carbon cycle to global warming. We include those uncertainties in our analysis to derive probabilities of triggering for 16 previously-identified climate tipping points within the Earth system. To conduct our analysis, we use the intermediate complexity climate model FaIR which is coupled to a conceptual model of the tipping processes within the Amazon rainforest and permafrost, which are the two major tipping elements within the Earth's carbon cycle. Uncertainties are propagated by employing a Monte Carlo approach for the construction of large model ensembles. We find that intermediate emission scenarios like SSP2-4.5 are highly unsafe with regard to triggering climate tipping points, with an average probability of triggering until the year 2500 of 65%. Furthermore, the highest long-term temperature increase among all SSPs caused by carbon emissions from the Amazon and permafrost becomes possible under this scenario with $0.16°C$ $(0.03 - 0.91°C)$ in 2500, which increases the average probability of triggering tipping points by 3.3 percent points (pp). This is due to the fact that maximum carbon emissions from tipping of the Amazon and permafrost become possible under this scenario, and they cause most warming when cumulative anthropogenic emissions are lower due to the saturating response of radiative forcing to increasing greenhouse gas concentrations. The risk of triggering climate tipping points is reduced significantly under SSP1-2.6 and even more so under SSP1-1.9, with average probabilities of triggering of 38% and 28% respectively, which are increased by 2.3 pp and 1.1 pp due to carbon emissions from the Amazon and permafrost.



# 1 Introduction

The term "tipping point" is commonly used to describe a critical threshold in the forcing of a system, at which small additional forcing leads to significant and long-term changes of the system (Lenton et al., 2008). The debate about tipping points in the climate system, referred to as "climate tipping points", has intensified over the past two decades. Increasing numbers of Earth system components have been identified which could possibly exhibit tipping behaviour (e.g. Lenton et al., 2008; Kriegler et al., 2009; Steffen et al., 2018), with 15 candidates being shortlisted in the latest IPCC report (Lee et al., 2021). These Earth system components are referred to as "tipping elements" (TEs) and occur within the biosphere, cryosphere and oceanic or atmospheric circulation (Lenton et al., 2008). The global mean surface temperature (GMST) relative to pre-industrial levels is used as a common metric to describe the forcing of the TEs (Armstrong McKay et al., 2022). This means that a tipping point can be associated with a "threshold temperature" after which the respective TE is expected to exhibit tipping behaviour. There is growing concern about the possible proximity of climate tipping points, as threshold temperatures have been revised to lower levels, with some TEs being at risk of getting "triggered" (crossing of their threshold temperature) at GMST values as low as $1°C$ (Lenton et al., 2019).

In their recent literature synthesis, Armstrong McKay et al. (2022) identify 16 TEs within the Earth system and provide estimates of their threshold temperatures, the characteristic timescales their tipping is assumed to unfold over, and their impact on global warming (Table 1, we will use the abbreviations defined in the table for the different TEs). Building on the definition of TEs by Lenton et al. (2008), Armstrong McKay et al. (2022) distinguish between "global core" and "regional impact" TEs. To qualify as a global core TE, a tipping point has to occur uniformly across a sub-continental scale ($\sim 1000\,\mathrm{km}$) for a subsystem of the Earth. However, if the change in forcing is approximately uniform across a large spatial area, small-scale tipping points can be crossed near-synchronously at a sub-continental scale, which qualifies the affected subsystem as a regional impact TE. Furthermore, global TEs have to contribute significantly to the overall operation mode of the Earth system, while regional impact TEs are required to either contribute significantly to human welfare or to have great value in themselves as unique features of the Earth system.

Since triggering of TEs will negatively affect human welfare, political efforts should be increased to avoid them (Cai et al., 2016). However, it is not straightforward to determine how safe a specific emission scenario is with regard to triggering TEs, since several uncertainties need to be accounted for within this calculation. The climate sensitivity to anthropogenic emissions remains poorly constrained, with a likely range of the equilibrium climate sensitivity to a doubling of atmospheric $CO_2$ concentrations of $2.5 - 4°C$ (Chen et al., 2021). Furthermore, climate tipping points include uncertainties within their threshold temperatures, timescales, impacts and for some even their existence remains uncertain (Armstrong McKay et al., 2022). Additional uncertainty is introduced by potential interactions of climate tipping points, which tend to destabilize them (Wunderling et al., 2021). Such interactions can be of manifold nature and often involve complex mechanisms. One example are TEs within the Earth's carbon cycle, which have the potential to release large amounts of greenhouse gases (GHGs) and thereby amplify global warming, which in turn increases the probability of triggering other TEs (Steffen et al., 2018).



In this study, we calculate probabilities of triggering for the 16 TEs that Armstrong McKay et al. (2022) identify within the
Earth system, including the uncertainties in climate sensitivities and the threshold temperatures, by employing a Monte Carlo
approach (Metropolis et al., 1953). Herewith, we provide an update of the widely used but somewhat outdated probabilities
of triggering derived from an expert elicitation conducted by Kriegler et al. (2009). Furthermore, we quantify the additional
warming that might arise from TEs within the Earth's carbon cycle and how it increases the probabilities of triggering other
TEs.

TEs with the potential to significantly impact the Earth's carbon cycle include abrupt permafrost thaw or collapse, Amazon
rainforest dieback and northern expansion and southern dieback of boreal forest (Armstrong McKay et al., 2022). Since north-
ern expansion and southern dieback of boreal forests are assumed to roughly balance out in terms of global warming (Canadell
et al., 2021), we focus on permafrost thaw and Amazon dieback. Permafrost thaw can be further subdivided into three distinct
processes: gradual thaw, a threshold-free feedback to global warming, abrupt thaw, a regional impact tipping element, and
collapse, a global core tipping element (Armstrong McKay et al., 2022). Since gradual thaw of permafrost, associated with
uniform and large-scale deepening of the active layer, is not assumed to include a tipping point, we do not include it in our
analysis, instead focusing on abrupt thaw and collapse of permafrost and Amazon dieback, hereafter referred to as the "carbon
tipping elements".

Abrupt thaw of permafrost occurs regionally but near-synchronously over the permafrost region due to thermokarsts which
can affect several meters of permafrost within days to weeks (Turetsky et al., 2019). The emerging landscapes mostly include
water-saturated soils (Olefeldt et al., 2016), hence high methane emissions of around 20% from anaerobic respiration of the
now accessible soil organic carbon can be expected (Turetsky et al., 2020).

Collapse of permafrost can be caused by permafrost degradation becoming self-perpetuating due to the heat released by micro-
bial respiration of soil organic carbon, leading to further thaw of permafrost and resulting in a positive feedback loop, which is
referred to as the "compost bomb instability" (Luke and Cox, 2011; Khvorostyanov et al., 2008). This process might occur in
the Yedoma region or in abruptly dried permafrost soils (Armstrong McKay et al., 2022).

Abrupt dieback of the Amazon is assumed to occur due to reduced moisture recycling and forest-fire-feedbacks triggered by
initial tree loss due to either global warming or deforestation, whereby only the former is accounted for in this study (Science
Panel for the Amazon, 2021; Nobre et al., 2016).

Our model framework relies on the second version of the Finite amplitude Impulse Response model (FaIR), a 0D reduced
complexity climate model developed by Leach et al. (2021), and the estimates for the threshold temperatures, timescales, and
impacts of TEs from Armstrong McKay et al. (2022). These estimates are used to build a conceptual carbon tipping elements
model (CTEM), which can be coupled to FaIR to include the additional carbon emissions from carbon TEs. With this setup,
we generate a "coupled" (CTEM coupled to FaIR) and an "uncoupled" (FaIR only) large-scale model ensemble for all Tier1
shared socioeconomic pathways (SSPs) (O'Neill et al., 2016), propagating the involved uncertainties, up until the year 2500.
Even though we give results for all SSPs, we focus on SSP2-4.5 and SSP1-2.6. The "middle of the road" scenario SSP2-4.5 is
the scenario current policies and actions are roughly following, with an estimated warming of 2.7°C in 2100 (Climate Action
Tracker, 2022). Increased political efforts that would guarantee the timely and complete fulfilment of current climate change





mitigation pledges can still limit global warming to 2°C (Meinshausen et al., 2022), consistent with SSP1-2.6. Since SSP1-1.9
might already be politically infeasible to achieve (Jewell and Cherp, 2020), SSP1-2.6 constitutes a more realistic "best-case"
scenario. To get a sense of how much safer the world would be from crossing climate tipping points when moving away
from the current middle of the road towards a more sustainable pathway, it is therefore interesting to compare SSP2-4.5 and
SSP1-2.6.

Our approach is explained in more detail in the next section. In section 3 we investigate the carbon emissions and their
additional warming from carbon TEs, followed by a presentation of the probabilities of triggering TEs in section 4. We discuss
of our results in section 5 and conclude with section 6.

## 2   Data and Methods

### 2.1   SSP Emission Pathways

We use the CMIP6 GHG and aerosol emissions and effective radiative forcing datasets employed in the Reduced Complexity
Model Intercomparison Project (Nicholls et al., 2020). Both datasets contain global average values with an annual timestep
throughout the historical period (1750-2014) which shifts to a decadal timestep for the projection period (2015-2500).

The CMIP6 GHG and aerosol emission projections for the five different SSP scenarios follow Gidden et al. (2019). The emis-
sion extensions beyond 2100 follow the conventions described in Meinshausen et al. (2020). Historical emissions (1750–2014)
of chemically reactive gases (CO, $CH_4$, $NH_3$, $NO_x$, $SO_2$, non-methane volatile organic compounds) carbonaceous aerosols
(black carbon and organic carbon) and $CO_2$ come from the Community Emissions Data System (Hoesly et al., 2018). Histori-
cal biomass burning emissions of $CH_4$, black carbon, CO, $NH_3$, $NO_X$, organic carbon, $SO_2$ and non-methane volatile organic
compounds come from Van Marle et al. (2017). Global historical $CO_2$ emissions from land-use are taken from the Global
Carbon Budget 2016 (Le Quéré et al., 2016). Regional breakdown of land-use $CO_2$ emissions and $N_2O$ emissions come from
the Potsdam Real-time Integrated Model for probabilistic Assessment of emissions Paths for historical emissions version 1.0
(Guetschow et al., 2016). Data gaps in the historical emissions were filled with inverse emissions based on CMIP6 concentra-
tions from the Model for the Assessment of Greenhouse Gas Induced Climate Change (MAGICC) 7.0.0 (Meinshausen et al.,
2020).

Effective radiative forcing for CMIP6 follows the data provided by Smith (2020).

### 2.2   The FaIR Model

To map GHG and aerosol emissions to GMST, including the uncertainty in ECS, we use the second version of the FaIR
model developed by Leach et al. (2021), which is run with an annual timestep. It consists of six equations, five of which
are adopted from Myhre et al. (2013). The sixth equation implements a state-dependency of the carbon cycle, which enables
a better representation of the relationship between emissions and atmospheric concentrations for historical observations and
projections (Leach et al., 2021). Despite its relative simplicity, FaIR is flexible enough to emulate more complex ESMs from



| Category | Proposed Climate Tipping Element (& Tipping Point) | | Threshold (°C) | | | Timescale (years) | | | Max. Impact (GtC) |
|---|---|---|---|---|---|---|---|---|---|
| | | | Est. | Min | Max | Est. | Min | Max | |
| Global core tipping elements | PFTP | Boreal Permafrost (collapse) | 4.0 | 3.0 | 6.0 | 50 | 10 | 300 | 125 - 250 |
| | GrIS | Greenland Ice Sheet (collapse) | 1.5 | 0.8 | 3.0 | 10k | 1k | 15k | |
| | WAIS | West Antarctic Ice Sheet (collapse) | 1.5 | 1.0 | 3.0 | 2k | 500 | 13k | |
| | EAIS | East Antarctic Ice Sheet (collapse) | 7.5 | 5.0 | 10.0 | ? | 10k | ? | |
| | EASB | East Antarctic Subglacial Basin (collapse) | 3.0 | 2.0 | 6.0 | 2k | 500 | 10k | |
| | AWSI | Arctic Winter Sea Ice (collapse) | 6.3 | 4.5 | 8.7 | 20 | 10 | 100 | |
| | AMAZ | Amazon Rainforest (dieback) | 3.5 | 2.0 | 6.0 | 100 | 50 | 200 | Partial: 30 Total: 75 |
| | AMOC | Atlantic M. O. Circulation (collapse) | 4.0 | 1.4 | 8.0 | 50 | 15 | 300 | |
| | LABC | Labrador- Irminger Seas SPG Convection (collapse) | 1.8 | 1.1 | 3.8 | 10 | 5 | 50 | |
| Regional impact tipping elements | PFAT | Boreal Permafrost (abrupt thaw) | 1.5 | 1.0 | 2.3 | 200 | 100 | 300 | Adds 25-75% to gradual thaw: = 5-15 GtC per °C @2100; 12.5-37.5 GtC per °C @2300, up to max. 60-195 GtC |
| | BARI | Barents Sea Ice (abrupt loss) | 1.6 | 1.5 | 1.7 | 25 | ? | ? | |
| | GLCR | Mountain Glaciers (loss) | 2.0 | 1.5 | 3.0 | 200 | 50 | 1k | |
| | REEF | Low-latitude Coral Reefs (die-off) | 1.5 | 1.0 | 2.0 | 10 | - | - | |
| | SAHL | Sahel and W. African Monsoon (greening) | 2.8 | 2.0 | 3.5 | 50 | 10 | 500 | |
| | BORF | Boreal Forest (southern dieback) | 4.0 | 1.4 | 5.0 | 100 | 50 | ? | 52 GtC / -0.18°C |
| | TUND | Boreal Forest (northern expansion) | 4.0 | 1.5 | 7.2 | 100 | 40 | ? | -6 GtC / +0.14°C |
| Threshold-free nonlinear Feedbacks | PFGT | Boreal Permafrost (gradual thaw) | 1.5 | 1.0 | 2.4 | 300 | 100 | 300< | 20 GtC per °C @2100; 50 GtC per °C @2300, up to max 260 GtC |

**Table 1.** Category, threshold, timescale, and impact of the TEs, reproduced from Armstrong McKay et al. (2022). Colours of the second left column represent the Earth system domain of the tipping point (blue = cryosphere, green = biosphere, orange = ocean/atmosphere). All other colours represent confidence, with green = high confidence, yellow = medium confidence and red = low confidence.



CMIP6. FaIR allows for probabilistic projections of GMST by relying on a parameter ensemble informed by CMIP6 models and constrained with the observed trend and level of global warming. However, this constraint is not strong enough to rule out potentially unrealistic high equilibrium climate sensitivities (ECS) with a 95th percentile of 6.59°C, compared to 5°C reported by Forster et al. (2021). Since we rely on this constrained parameter ensemble to include the uncertainty in ECS in our predictions of GMST, extremely high temperatures towards the end of the model period might be unrealistic.

### 2.3 Carbon Tipping Elements Model

We introduce the simple, system-dynamics type model CTEM to investigate the carbon emissions that might arise from tipping of AMAZ, PFTP and PFAT and the respective warming caused by them, which, in turn, increases the probabilities of triggering other TEs. CTEM is able to represent the carbon emissions from AMAZ, PFAT, and PFTP, with threshold temperatures, timescales, and impacts consistent with the estimates from Armstrong McKay et al. (2022) (Table 1). Each TE is represented by a stock of cumulative carbon emissions it adds to the SSP carbon emissions. The cumulative emissions are represented by the following logistic equation:

$$\frac{dS}{dt} = r\left(\frac{T}{P}\right)S\left(1 - \frac{S}{K}\right), \tag{1}$$

with $S$ being cumulative carbon emissions (in GtC), $r$ the maximum growth rate (in $\mathrm{yr}^{-1}$), $K$ the maximum impact (in GtC), $P$ the threshold temperature (in °C), $T$ GMST relative to pre-industrial levels (in °C) and $t$ time (in years). The rate-dependence of all three TEs found by Armstrong McKay et al. (2022) is included by the term $T/P$, which means higher exceedance of the threshold temperature causes faster change in $S$.

For AMAZ and PFTP, the impact is independent of GMST once they are triggered (Table 1). Therefore, Eq. 1 can be used to represent them without further modifications being necessary. PFAT, however, is assumed to amplify PFGT, which is a threshold-free feedback. Therefore, the impact of PFAT also depends on GMST, with higher temperatures leading to increased carbon emissions, and different values for this feedback in 2100 and 2300 (Table 1). Hence, for PFAT $K$ from Eq. 1 needs to be calculated as

$$K = \min(F \cdot T, \, K_{\max}) \tag{2}$$

with

$$F = \begin{cases} F_{100} & \forall \, t \leq 2100 \\ F_{100} \cdot \frac{2300-t}{200} + F_{300} \cdot \frac{t-2100}{200} & \forall \, 2100 < t < 2300 \\ F_{300} & \forall \, t \geq 2300 \end{cases} \tag{3}$$

Here, $K$ is calculated as the product of the feedback strength $F$ (in GtC °C$^{-1}$) and GMST (in °C), limited to the maximum impact $K_{\max}$ (in GtC) of PFAT. In Eq. 3, we define $F$ as the feedback strength for the year 2100 ($F_{100}$) before and in 2100, as the feedback strength for the year 2300 ($F_{300}$) in and after 2300 and interpolate linearly between these two values between 2100 and 2300.





The resulting set of three differential equations (one for each TE) is solved using a forward difference scheme, similar to the
implementation of FaIR (supplement of Leach et al. (2021)). In FaIR, output variables such as $T$ are assumed to be average
values between two consecutive timesteps (denoted by a bar over $t$), while the values for the input variables such as the annual
SSP GHG emissions ($E_{\mathrm{SSP}}$) reside at each timestep (no bar over $t$). To be consistent with this implementation, we define $S$ and
the resulting annual carbon emissions from each TE ($E_{\mathrm{TE}}$) also at each timestep (Fig. S1). To calculate $S$ for each timestep ($t$)
with a step size ($\Delta t$) of one year, we integrate Eq. 1 which yields:

$$S(t) = \left( e^{-a(\overline{t-1})\Delta t} \left( S^{-1}(t-1) - \frac{b(\overline{t-1})}{a(\overline{t-1})} \right) + \frac{b(\overline{t-1})}{a(\overline{t-1})} \right)^{-1}, \tag{4}$$

with the auxiliary variables $a$ and $b$ being calculated as

$$a(\overline{t-1}) = r\frac{T(\overline{t-1})}{P}$$
$$b(\overline{t-1}) = \frac{rT(\overline{t-1})}{PK}.$$

For PFTP and AMAZ Eq. 4 can be used directly to calculate $S$, but for PFAT $K$ needs to be calculated using Eq. 2 with $T = T(\overline{t-1})$ and is therefore also time-dependent (for the full derivation please see Sect.S1). Irreversibility of carbon emissions
from all three TEs is implemented by setting negative changes of $S$ to zero.

CTEM needs to be calibrated to match the estimated behaviour summarised in Table 1. While values for $F_{100}$, $F_{300}$, $K$,
$K_{\max}$, and $P$ can be used in CTEM directly, $r$ and the initial stock ($S_0$) need to be calibrated or defined for the model to
match the proposed tipping timescales ($H$). For this purpose, we define the $H$ as the period over which 99% of the cumulative
emissions occur. While $H$ and the corresponding value of $r$ are calibrated for the whole range of $H$ given in Table 1 for PFTP
and AMAZ, PFAT is implemented slightly different since $K$ depends on $T$ (Eq. 2). For PFAT, $H$ is included implicitly in
the feedback parameters $F_{100}$ and $F_{300}$, with high values corresponding to short $H$ and vice versa. Therefore, we keep $H$
and the corresponding $r$ fixed at their mean value for PFAT, with varying values for the feedback parameters. In the calibrated
version of CTEM, the evolution of $S$ follows the characteristic S-shape of logistic equations (Fig. 1) Further information on the
calibration of CTEM is given in Sect.S2. A test of the calibration yields that the timescales produced by CTEM are generally
matching the expected behaviour for AMAZ and PFTP (Fig. S3), but not for PFAT (Fig. S4). For PFAT we find that the carbon
emissions are generally too low in 2100, however, this emission deficit is removed until 2300, where emissions of PFAT match
the estimates. We regard this deviation to be acceptable, given the simplicity of our model approach (see Sect.S3 for further
information).

### 2.3.1 Coupling to FaIR

CTEM is coupled to FaIR every timestep by adding $E_{\mathrm{TE}}$ to $E_{\mathrm{SSP}}$. For each TE, we calculate $E_{\mathrm{TE}}$ from $S$ as

$$E_{\mathrm{TE}}(t) = S(t) - S(t-1).$$

The total carbon emissions $E_{\mathrm{TE}}$ are split up into $CO_2$ and $CH_4$ emissions for all three TEs. For PFAT, we assume that 20% of
the carbon is emitted as $CH_4$, following Turetsky et al. (2020), while no $CH_4$ emissions are expected from AMAZ (Armstrong



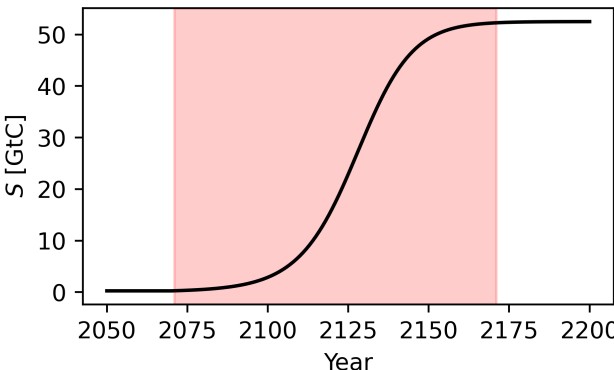

**Figure 1.** $S$ for the calibration of the mean timescale of AMAZ (under SSP5-8.5). The red shaded area denotes the period over which 99% of the cumulative carbon emissions occur.

McKay et al., 2022). Quantifying the fraction of $CH_4$ emissions arising from PFTP is challenging due to the lack of previous model studies and limited process understanding. We rely on the 2D model study conducted by Schneider von Deimling et al. (2015), who report 40% of additional warming due to $CH_4$ emissions from the Yedoma deposits, which can be associated with PFTP. Since this amplification is within the uncertainty range of additional warming caused by $CH_4$ emissions from gradual permafrost thaw of $35 - 48\%$ for which 2.3% of the carbon is emitted as methane (Schuur et al., 2015), we assume that this

fraction of $CH_4$ emissions also holds for PFTP. The annual $CH_4$ and $CO_2$ emissions of all three TEs are then added to $E_{SSP}(t)$ of the respective SSP and the sum is used to run the next timestep of FaIR, calculating $T(\bar{t})$ which is then used to force CTEM and so on.

## 2.4   Generation of Model Ensembles

We construct a coupled (FaIR coupled to CTEM) and an uncoupled (FaIR only) 5000 members model ensemble for all five
Tier1 SSPs. Hereby, we follow a Monte Carlo scheme (Metropolis et al., 1953) to propagate the uncertainties within FaIR and CTEM.

For the parameterization of FaIR, we randomly select 5000 members of its constrained parameter ensemble. We use the same parameter sample for FaIR in both the coupled and the uncoupled ensemble to make sure that changes in the climate response only arise from carbon emissions of the carbon TEs.

For the parameterization of CTEM and the calculation of triggering probabilities, we construct probability distributions of the respective parameters based on the estimates given in Table 1. From those probability distributions, we sample parameter values, employing Latin hypercube sampling (McKay et al., 1979).

To calculate the probability of triggering, we sample the uncertainty range of $P$ for all TEs, again using the same parameter sample for both ensembles. For the sampling process of $P$, we infer probability distributions from the estimates in Table 1,





**Figure 2.** Cumulative distribution functions of $P$ for all TEs, together with the RMSE between the given percentiles (red dots) and the actual percentiles of the respective distribution. The title states TE, distribution type and RMSE in °C.



deciding that the minimum estimate, the best estimate, and the maximum estimate should correspond to the 5th, the 50th and the 95th percentile of the respective probability distribution. To determine the probability distributions which best fits those percentiles, we use the "rriskDistributions" package (Belgorodski et al., 2017), which chooses out of 17 continuous probability distributions, minimizing the absolute difference between the given and the actual percentiles. We derive eight log-normal, four triangular, three normal and one Gompertz distribution (Fig. 2). While some distributions of $P$ agree perfectly with the given percentiles (AMOC, EAIS, TUND, BARI, SAHL), others deviate substantially with RMSEs higher than 0.2°C (WAIS, BORF, EASB, LABC). In the case of WAIS, EASB, and LABC this deviation is mainly caused by too low values at the 95th percentile, which is reached at 2.3°C rather than 3°C for WAIS, 4.6°C rather than 6°C for EASB, and 3.1°C rather than 3.8°C for LABC. In the case of BORF, the relatively high RMSE is caused by a too high value at the 5th percentile (1.7°C rather than 1.4°C) and a too low value at the 50th percentile (3.8°C rather than 4°C), while the 95th percentile agrees well with the expected value (less than 0.02°C deviation). Even though the deviations between the given and the actual percentiles of those four TEs are significant, we regard them to be acceptable, given that they cannot be reduced with our methodology, meaning that other probability distributions would be needed to achieve this. However, it must be kept in mind that high probabilities of tipping might be reached at too low values of GMST for WAIS, EASB and LABC due to their imperfect probability distributions of $P$.

We now turn to the parameterization of CTEM. For the distribution of $H$ for PFTP and AMAZ, we derive one log-normal distribution each, following the same approach as for the generation of probability distributions of $P$ (Fig. S5). The impacts $K$ of PFTP and AMAZ and the maximum impact $K_{\max}$ of PFAT are sampled from continuous uniform distributions, with the same probability for all values within the given ranges from Table 1 (Fig. S6). We regard this reasonable since Armstrong McKay et al. (2022) only give maximum and minimum values for those variables, and we do not have any additional information about their distribution. The feedback strengths $F_{100}$ and $F_{300}$ of PFAT are also sampled from continuous uniform distributions, with the same probability for all values within the given ranges from Table 1 (Fig. S7). Here, the same argument holds as for the selection of the distributions of $K$ and $K_{\max}$.

We assume that all parameters within the parameter set of CTEM are uncorrelated, except for $F_{100}$ and $F_{300}$, which are correlated with a correlation coefficient of 1. The decision about the correlation of the parameters represents our understanding that there are no correlations, or at least no clear evidence for them, between the threshold temperatures, the timescales, or the impacts of AMAZ, PFAT and PFTP. The correlation between $F_{100}$ and $F_{300}$ is established since $H$ of PFAT is included in those parameters, and we assume that $H$ is constant over time, e.g., a low feedback in 2100 associated with a high $H$ also means low feedback in 2300 due to the same high $H$ the emissions occur over.

## 2.5 Calculation of Tipping Probabilities

To calculate the probabilities of triggering any of the 16 TEs discussed in this study, we sample one value of $P$ from the respective distribution (Fig. 2) for each TE and each ensemble member. If $T$ of that ensemble member exceeds $P$, the TE is counted as triggered. In this manner, we derive the share of ensemble members in which a TE is triggered for each TE at all times. As all TEs discussed here are by definition irreversible on the considered timescale, the share of triggered TEs cannot





decrease with time, even if $T$ would decrease. The share of ensemble members with triggered TEs is then used to further
investigate the probabilities of triggering, e.g., by calculating the timing of the exceedance of certain share percentages.

## 2.6    Robustness of the Results

To examine whether an ensemble of size 5000 is sufficiently large to approximate the distributions of the output variables
with our Monte Carlo approach, we create 20 coupled ensembles with 5000 members and run under SSP5-8.5, sampling
uncertainties within FaIR and CTEM. As carbon emissions from CTEM are used to calculate $T$, which is also the final output
variable of FaIR, it includes uncertainties from all parameters and is hence expected to vary most between different ensembles.
Therefore, we inspect the deviation of $T$ between the 20 ensembles. While the 5th percentile and the mean of $T$ are nearly
equal for all ensembles, with standard deviations below 0.06°C at all times, the 95th percentile deviates slightly between the
ensembles, with a standard deviation of up to 0.24°C (Fig. S8). Given the range of $T$ between the 5th and the 95th percentile
of ∼ 11°C, we regard those deviations to be sufficiently small.

## 3    Impacts of Carbon Tipping Elements

We now turn to the presentation of our results. For the description of probabilities we use the calibrated language of the IPCC:
virtually certain: 99 – 100%, extremely likely: 95 – 100%, very likely: 90 – 100%, likely: 66 – 100%, more likely than not: >50
– 100%, about as likely as not: 33 – 66%, unlikely: 0 – 33%, very unlikely: 0 – 10%, and extremely unlikely: 0 – 5% (Chen
et al., 2021).

## 3.1    Additional Carbon Emissions

The carbon emissions from carbon TEs increase from SSP1-1.9 to SSP5-8.5 (Table 2, Fig. S9). While zero carbon emissions
from TEs remain possible under SSP1-1.9 and SSP1-2.6, high carbon emissions become much more likely under SSP2-4.5,
with the maximum emissions being nearly as high as under SSP5-8.5 (Table 2). This is a direct consequence of the probabilities
of triggering carbon TEs. Under SSP1-2.6 only triggering of PFAT becomes likely in the coupled ensemble (82%), while
triggering of AMAZ and PFTP remains unlikely (13% and 5% respectively). Under SSP2-4.5, triggering probabilities are
increased substantially, to 98% for PFAT, 55% for AMAZ and 38% for PFTP. This means that maximum carbon emissions
with all three carbon TEs triggered become more likely under SSP2-4.5 compared to SSP1-2.6.

Even though the highest carbon emissions from carbon TEs are emitted under SSP3-7.0 and SSP5-8.5, they remain small
compared to anthropogenic carbon emissions (Fig. 3). The 95th percentile of cumulative carbon emissions from carbon TEs
only reaches 14% of the cumulative anthropogenic carbon emissions in 2500 under SSP3-7.0, and 9% under SSP5-8.5. The
relative contribution of carbon emissions from carbon TEs to the total carbon emissions increases towards lower emission
scenarios. Under SSP2-4.5 the 95th percentile of cumulative carbon emissions from carbon TEs reaches 35% of the cumulative
anthropogenic carbon emissions in 2500, 40% under SSP1-2.6 and 84% under SSP1-1.9.





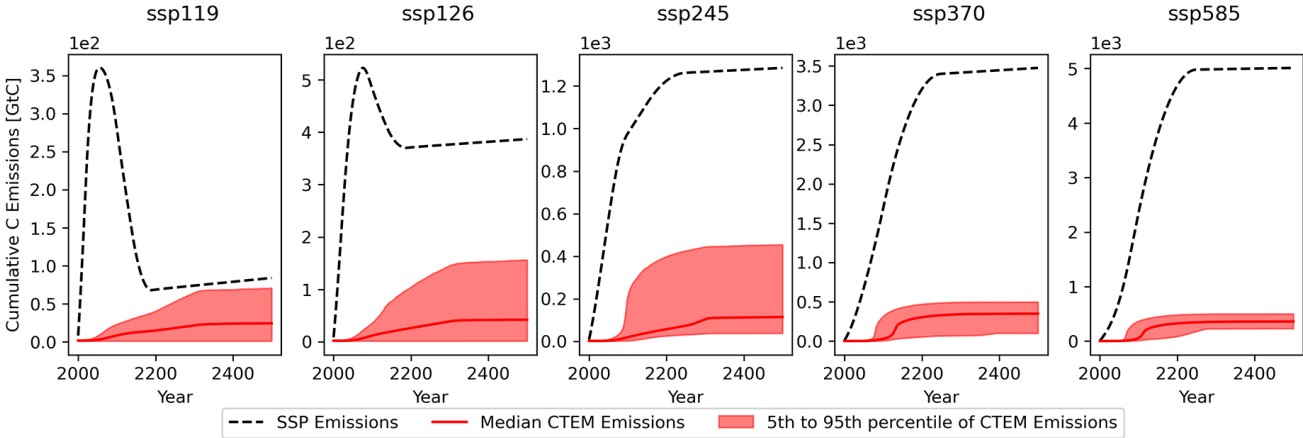

**Figure 3.** Carbon emissions and cumulative carbon emissions from the Tier1 SSP scenarios and the carbon TEs (modelled by CTEM) between 2000 and 2500.

Hence, SSP2-4.5 is special in the sense that maximum carbon emissions from carbon TEs are already possible and are large relative to the anthropogenic emissions.

### 3.2 Increase of Global Warming

The carbon emissions from carbon TEs cause an increase in global warming (Fig. 4). Under the low emission scenarios SSP1-1.9 and SSP1-2.6, high temperature increases from carbon TEs are possible but less common, with the temperature distributions skewed towards higher values. No temperature increase from carbon TEs remains possible under those scenarios, as it is still within the very likely range (Table 2). The median temperature increase from carbon TEs also remains relatively small, reaching maximum values of 0.05°C and 0.1°C in 2300 under SSP1-1.9 and SSP1-2.6 respectively. However, the upper end of the very likely range of the temperature increase from carbon TEs reaches maxima of 0.29°C in 2300 under SSP1-1.9 and 0.53°C in 2500 under SSP1-2.6, which would mean a 26% and 32% temperature increase respectively compared to the median of the uncoupled ensemble.

For SSP2-4.5 high impacts from carbon TEs remain an exception (Fig. 4), however, compared to the low emission scenarios high temperature increases become more common, as reflected by the median temperature increase, which reaches its maximum of 0.22°C in 2300 (Table 2). Even though the warming caused by anthropogenic carbon emissions has increased under SSP2-4.5, the additional warming caused by carbon TEs can still cause high relative temperature increases. In 2500 the upper end of the very likely range of additional warming from carbon TEs reaches 0.91°C (Table 2), which constitutes a temperature increase of 29% compared to the median of the uncoupled ensemble.

Additional warming from carbon TEs becomes the default case under the high emission scenarios SSP3-7.0 and SSP5-8.5. However, this additional warming becomes small compared to the warming from anthropogenic carbon emissions. In 2500





|  |  | 2050 | 2100 | 2200 | 2300 | 2400 | 2500 |
|---|---|---|---|---|---|---|---|
| **SSP1-1.9** | | | | | | | |
| $CO_2$ | [GtC] | 1 (0-5) | 5 (0-19) | 11 (0-36) | 16 (0-58) | 18 (0-63) | 18 (0-66) |
| $CH_4$ | [$GtCH_4$] | 0.3 (0-1.5) | 1.8 (0-5.6) | 3.5 (0-10.2) | 5.3 (0-16.7) | 5.9 (0-17.8) | 6 (0-18.3) |
| $dT$ | [°C] | 0.01 (0-0.07) | 0.04 (0-0.17) | 0.03 (0-0.2) | 0.05 (0-0.29) | 0.03 (0-0.23) | 0.03 (0-0.22) |
| **SSP1-2.6** | | | | | | | |
| $CO_2$ | [GtC] | 1 (0-5) | 8 (0-29) | 20 (0-85) | 30 (0-121) | 33 (0-136) | 33 (0-152) |
| $CH_4$ | [$GtCH_4$] | 0.3 (0-1.6) | 2.6 (0-7.5) | 6.4 (0-15.9) | 9.8 (0-25.3) | 10.5 (0-27.0) | 10.7 (0-27.8) |
| $dT$ | [°C] | 0.01 (0-0.07) | 0.07 (0-0.24) | 0.08 (0-0.34) | 0.1 (0-0.49) | 0.06 (0-0.47) | 0.06 (0-0.53) |
| **SSP2-4.5** | | | | | | | |
| $CO_2$ | [GtC] | 1 (0-6) | 16 (2-155) | 67 (19-324) | 102 (28-359) | 108 (30-365) | 112 (30-369) |
| $CH_4$ | [$GtCH_4$] | 0.3 (0-1.8) | 4.7 (0.4-13.6) | 15.6 (6.3-33) | 21.9 (9.2-43.3) | 22.8 (9.6-44.7) | 23.1 (9.7-45.3) |
| $dT$ | [°C] | 0.01 (0-0.08) | 0.13 (0.02-0.61) | 0.2 (0.06-0.85) | 0.22 (0.06-0.91) | 0.16 (0.03-0.89) | 0.16(0.03-0.91) |
| **SSP3-7.0** | | | | | | | |
| $CO_2$ | [GtC] | 1 (0-6) | 34 (6-245) | 285 (91-381) | 318 (143-403) | 322 (154-406) | 324 (160-407) |
| $CH_4$ | [$GtCH_4$] | 0.4 (0-1.9) | 8.1 (1.5-20) | 29.3 (17-47.5) | 35.2 (22.9-53.9) | 35.6 (23.4-53.9) | 35.9 (23.6-54) |
| $dT$ | [°C] | 0.01 (0-0.08) | 0.25 (0.05-0.94) | 0.39 (0.15-0.81) | 0.31 (0.13-0.66) | 0.27 (0.1-0.63) | 0.26 (0.09-0.63) |
| **SSP5-8.5** | | | | | | | |
| $CO_2$ | [GtC] | 2 (0-8) | 78 (13-283) | 304 (138-389) | 328 (235-408) | 330 (244-410) | 331 (246-411) |
| $CH_4$ | [$GtCH_4$] | 0.5 (0-2.3) | 12.4 (3.3-26.2) | 32 (20.6-50.5) | 37.1 (24.1-54.3) | 37.4 (24.3-54.6) | 37.7 (24.3-54.7) |
| $dT$ | [°C] | 0.02 (0-0.1) | 0.4 (0.09-1.11) | 0.31 (0.13-0.64) | 0.23 (0.1-0.52) | 0.19 (0.08-0.47) | 0.18 (0.08-0.47) |

**Table 2.** Median and very likely range (5th – 95th percentile) of cumulative $CO_2$ and $CH_4$ emissions and the GMST increase ($dT$) caused by the carbon TEs.

under SSP5-8.5 the upper end of the very likely range of additional warming from carbon TEs reaches 0.47°C (Table 2), which only constitutes a temperature increase of 6% compared to the median of the uncoupled ensemble.

It is interesting to see that the highest long-term temperature increase from carbon TEs becomes possible under SSP2-4.5 and not under scenarios with higher anthropogenic emissions. This is in contradiction to the $CO_2$ and $CH_4$ emissions from carbon TEs, which are higher under high emission scenarios than under SSP2-4.5 (Table 2). The reason for this behaviour is the implementation of the forcing relationship translating atmospheric $CO_2$ and $CH_4$ concentrations to radiative forcing in FaIR. It is approximated by a logarithmic and a square-root term for $CO_2$ and by a square-root term for $CH_4$, meaning that the

effect of additional atmospheric concentrations of both GHGs on GMST is decreasing for higher atmospheric concentration levels. This means that the amount of carbon emissions from carbon TEs relative to the anthropogenic emissions is decisive to determine the additional warming from carbon TEs.





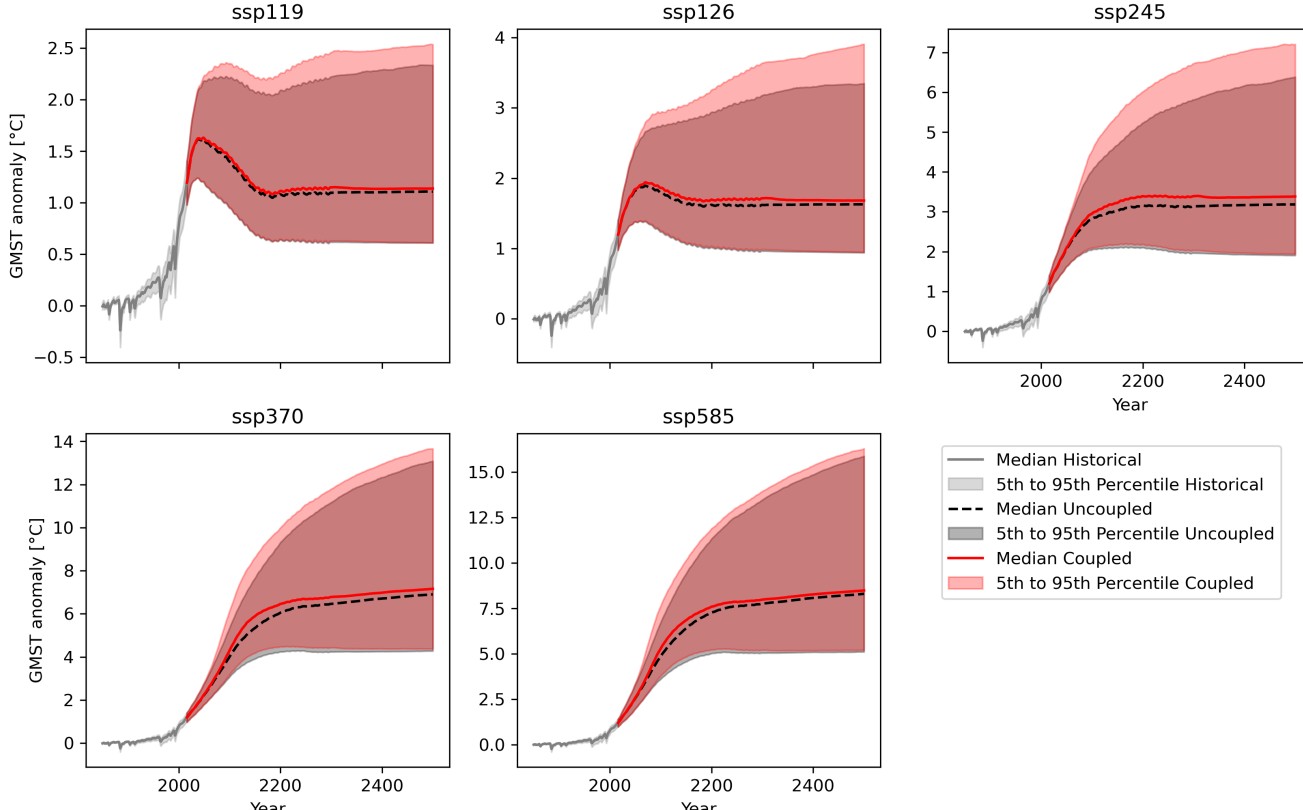

**Figure 4.** GMST relative to the 1850-1900 period of the coupled and the uncoupled ensemble for the Tier1 SSPs, together with the historical evolution.

Nevertheless, it must be noted that the highest short-term temperature increase from carbon TEs becomes possible under the high emissions scenarios SSP3-7.0 and SSP5-8.5, with the upper end of the very likely range reaching 0.94°C and 1.11°C in 2100 under SSP3-70 and SSP5-8.5 respectively. This can be linked to high transient $CH_4$ emissions from fast PFAT degradation due to rapidly increasing temperatures, which leads to peaks in the atmospheric $CH_4$ concentration anomaly around 2100 (Fig. S10).

The temperature increase from carbon TEs occurs earlier under high emission scenarios (Table 2), which can be explained by threshold temperatures of the carbon TEs being crossed earlier under those scenarios, as the general temperature increase is faster (Fig. 4). Interestingly, the median temperature increase from carbon TEs is declining towards the end of the model period under all SSPs (Table 2). This can be linked to the decreasing atmospheric methane concentrations (Fig. S10).







**Figure 5.** Cumulative number of TEs with a more than 50% probability (P) of getting triggered for the Tier1 SSP scenarios from the coupled and the uncoupled ensemble (upper row) and symbols of the triggered TEs corresponding to the increase of cumulative TEs in the coupled ensemble (lower row). Carbon TEs are marked with pink symbols, whereas the other TEs are marked in the colours suggested by Armstrong McKay et al. (2022), with blue for cryosphere, green for biosphere and orange for atmospheric or oceanic circulations.



## 4 Probabilities of Triggering Climate Tipping Elements

We now turn to the probabilities of triggering any of the 16 TEs proposed by Armstrong McKay et al. (2022) (Table 1), and how those probabilities are increased by the additional warming from carbon TEs. To compare those probabilities among the SSPs, we focus on the crossing of 50% probability, where triggering becomes more likely than not.

The effect of carbon TEs on the probabilities of triggering TEs under the different SSP scenarios is clearly visible, with triggering getting more likely earlier if carbon emissions from carbon TEs are included (Fig. 5). On average, TEs become more likely than not to be triggered 9.3 years earlier if carbon emissions from carbon TEs are included. This effect becomes more pronounced towards the end of the prediction period, since GMST is generally increasing slower at that time, which means it takes longer for the uncoupled ensemble to reach the same temperatures as the coupled ensemble (Fig. 4). Especially TEs within the cryosphere can become more likely than not to be triggered decades earlier, if carbon emissions from carbon TEs are included. EAIS and AWSI cross the 50% probability of triggering 25 years and 19 years earlier respectively under SSP5-8.5, AWSI crosses it 47 years earlier under SSP3-7.0, EASB crosses it 26 years earlier under SSP2-4.5, and GLCR crosses it 348 years earlier under SSP1-2.6 (Fig. 5). As we do not consider any interactions between the TEs apart from the increase in GMST from carbon TEs, the fact that crossing of a certain probability level is reached disproportionately earlier for some TEs if carbon emissions from carbon TEs are included is solely caused by a favourable combination of the threshold temperature distribution and the maximum temperatures reached under the respective SSP and should therefore not be over interpreted.

Even though additional probability of triggering caused by carbon emissions from carbon TEs clearly exists, it remains small compared to the probability of triggering caused by anthropogenic carbon emissions. Only two TEs become more likely than not to be triggered in addition to the uncoupled ensemble due to the inclusion of carbon TEs in the sum over all SSPs (Fig. 5). Carbon TEs do not cause high enough temperature increases to induce rapid triggering of multiple TEs following the tipping of the carbon TEs. Partly, this might be due to the fact that PFTP, which causes the highest carbon emissions of all carbon TEs, is one of the last TEs to be triggered and hence has little potential to cause earlier triggering of other TEs (Fig. 5). Moreover, even if all carbon TEs are extremely likely to be triggered, they only cause an increase of GMST of up to 0.4°C in the median under SSP5-8.5, which is too small to achieve crossing of multiple TE threshold temperatures (Table 2).

|  | Probability of Triggering [%] | Probability Increase from Carbon TEs [pp] |
| --- | --- | --- |
| SSP1-1.9 | 28 | 1.1 |
| SSP1-2.6 | 38 | 2.3 |
| SSP2-4.5 | 65 | 3.3 |
| SSP3-7.0 | 92 | 1.0 |
| SSP5-8.5 | 95 | 0.4 |

**Table 3.** Probability of triggering averaged over all TEs in 2500 and its increase due to additional warming caused by carbon emissions from carbon TEs in percent points (pp).



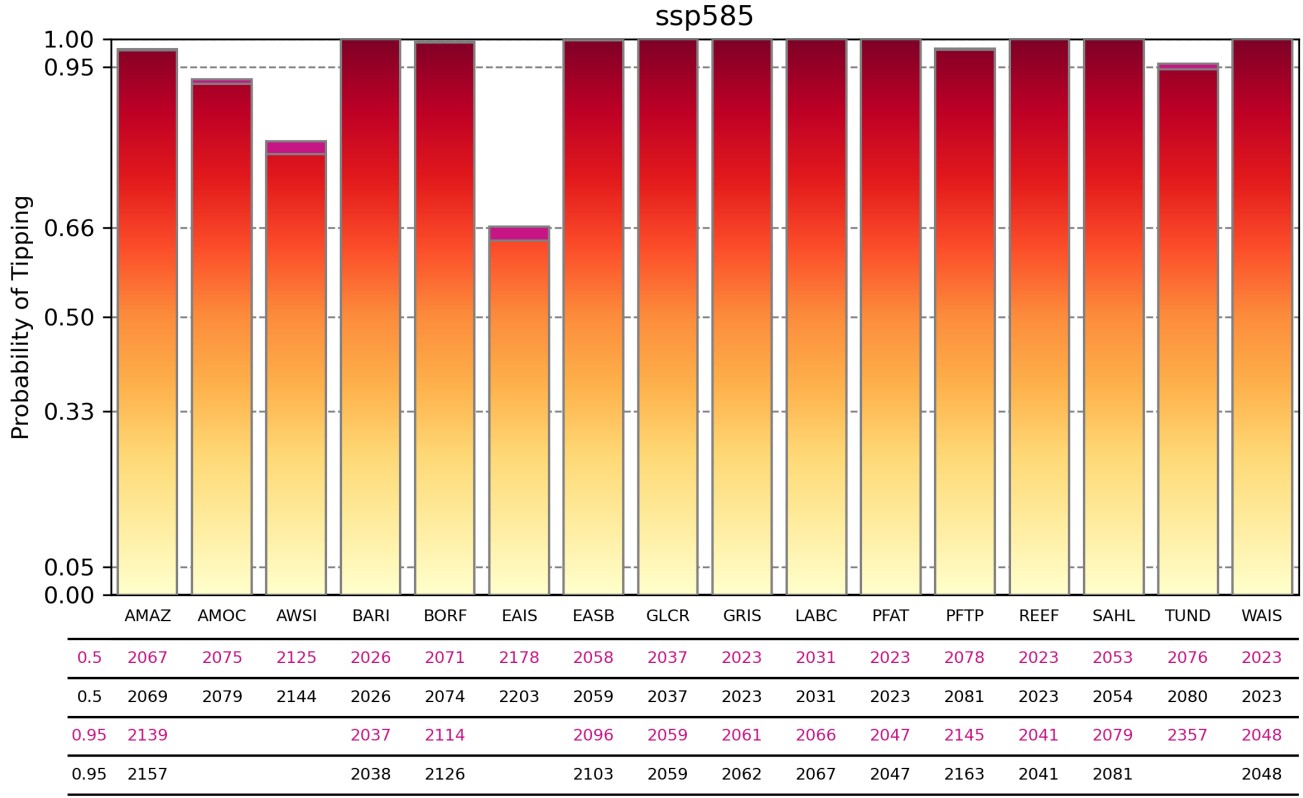

**Figure 6.** Probabilities of triggering the TEs by 2500 under SSP5-8.5. Additional probabilities from the carbon TEs are marked in purple. The table underneath states the years in which the 50% and 95% probability is crossed, with purple including carbon emissions from carbon TEs and black not.

For the exceedance of the 50% probability of triggering, we observe clustering of the TEs, which has the potential to occur earlier if carbon emissions from carbon TEs are included. The first cluster consisting of four TEs (GRIS, WAIS, REEF, and PFAT) become more likely than not to be triggered by 2026, regardless of the emission scenario or whether carbon emissions from carbon TEs are included or not (Fig. 5). Three additional TEs forming the second cluster (BARI, LABC and GLCR)
exceed the 50% probability of being triggered before 2050 under SSP2-4.5, SSP3-7.0 and SSP5-8.5. This is much delayed under SSP1-2.6, and can be partly avoided under SSP1-1.9 since only BARI becomes more likely to be triggered than not in 2031. The third cluster of seven TEs (SAHL, EASB, AMAZ, BORF, AMOC, TUND, and PFTP) crosses the 50% probability of triggering between 2050 and 2100 under SSP5-8.5 and SSP3-7.0. This can be partly avoided under SSP2-4.5 with only SAHL and EASB and if carbon emissions from carbon TEs are included also AMAZ and BORF becoming more likely than
not to be triggered until 2500. Under SSP1-1.9 and SSP1-2.6, none of the TEs from the third cluster becomes more likely to be triggered than not, independent of carbon emissions from the carbon TEs. The two remaining TEs AWSI and EAIS do also




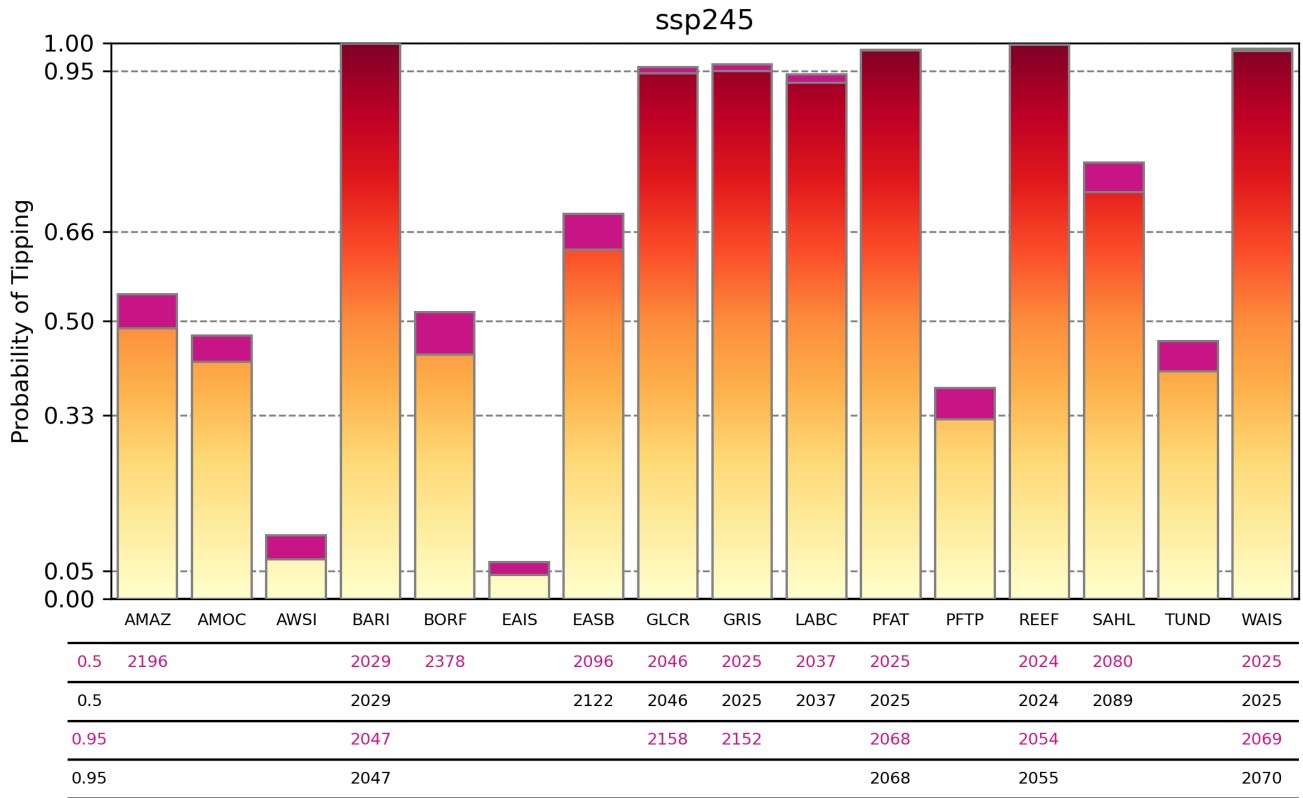

**Figure 7.** Probabilities of triggering the TEs by 2500 under SSP2-4.5. Additional probabilities from the carbon TEs are marked in purple. The table underneath states the years in which the 50% and the 95% probability are crossed, with purple including carbon emissions from carbon TEs and black not.

exceed the 50% probability of being triggered under SSP5-8.5 until 2300, while this is only the case for AWSI under SSP3-7.0. Hence, all 16 TEs are more likely than not to be triggered by 2500 under SSP5-8.5, 15 under SSP3-70, eleven (nine) under SSP2-4.5, seven under SSP1-2.6 and five under SSP1-1.9 including (not including) the carbon emissions from carbon TEs.

While the above inspection of the 50% probability level is especially valuable to understand the timing of triggering TEs, a more thorough analysis of the triggering probabilities is needed to fully understand how the risk of triggering TEs changes for the different SSPs. We turn to this part now. To compare the general risk of triggering TEs and how it is amplified by carbon TEs, we calculate average triggering probabilities at the end of our model period in 2500 (Table 3). Unsurprisingly, they increase from SSP1-1.9 to SSP5-8.5.

Under SSP5-8.5, triggering of most TEs becomes extremely likely and only the probability for triggering EAIS is remaining just below 66% (Fig. 6). The probabilities of triggering are reduced slightly under SSP3-7.0, but the general picture remains the same (Fig. S11). Interestingly, the additional probability of triggering TEs is lowest under the two high emission scenarios




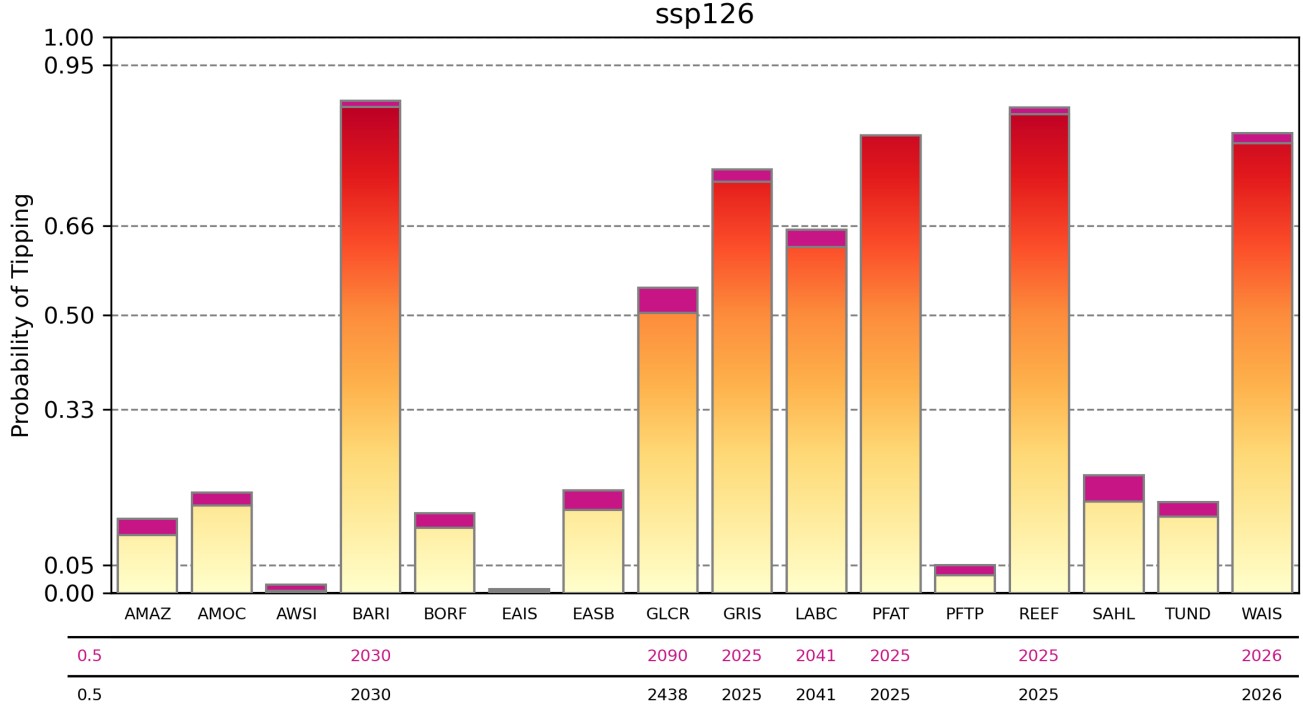

**Figure 8.** Probabilities of triggering the TEs by 2500 under SSP1-2.6. Additional probabilities from the carbon TEs are marked in purple. The table underneath states the years in which the 50% probability is crossed, with purple including carbon emissions from carbon TEs and black not.

(Table 3). Temperatures well above the threshold temperatures of most TEs are reached even without the effect of carbon TEs, hence, their impact on triggering probabilities is almost negligible.

Under SSP2-4.5 the probabilities of triggering vary strongly between different TEs. The four TEs BARI, REEF, PFAT, and WAIS are extremely likely to be triggered under SSP2-4.5, with GRIS and GLCR being extremely likely as well if carbon emissions from carbon TEs are included (Fig. 7). SAHL and, if carbon emissions from carbon TEs are included, EASB are likely to be triggered. Additional carbon emissions from carbon TEs also lead to AMAZ and BORF becoming more likely than not to be triggered. Only AWSI, EAIS, and, if carbon emissions from carbon TEs are not accounted, for PFTP remain unlikely

to be triggered, as TUND and AMOC are exceeding 33% probability of getting triggered in both ensembles. Triggering of EAIS remains extremely unlikely only if carbon emissions from the carbon TEs are not included.

    The increase in probabilities of triggering caused by carbon emissions from carbon TEs is highest under SSP2-4.5 (Table 3). This matches our finding that the highest long-term temperature increase from carbon TEs is also possible under this scenario (Table 2). Furthermore, GMST is generally increasing over time, which means that emissions from carbon TEs can produce

temperature maxima when they occur, which is not the case for lower emission scenarios (Fig. 4).





Moving from SSP2-4.5 to SSP1-2.6 reduces the probability of triggering multiple TEs significantly, namely by 28 pp (Table 3). No TE is extremely likely to be triggered under SSP1-2.6. However, the five TEs PFAT, REEF, GRIS, WAIS, and BARI are still likely and LABC and GLCR are more likely than not to be triggered, regardless of carbon emissions from carbon TEs (Fig. 8). All other TEs remain unlikely to be triggered, but only AWSI, EAIS and PFTP are extremely unlikely to be triggered under SSP1-2.6. The additional probability of triggering from carbon TEs is also reduced by one pp compared to SSP2-4.5 (Table 3).

SSP1-1.9 represents the safest scenario with respect to triggering TEs. However, the reduction of triggering probabilities compared to SSP1-2.6 is comparably small, with only 11.2 pp (Table 3). Even under SSP1-1.9 the five TEs REEF, WAIS, PFAT, GRIS and BARI become more likely than not to be triggered until 2031, regardless of the temperature impact from carbon TEs (Fig. S12). REEF, WAIS, PFAT, and, if carbon emissions from carbon TEs are included, BARI even become likely to be triggered. This means that if the assumed TE temperature threshold distributions from Armstrong McKay et al. (2022) are accurate, it is already too late to steer the Earth towards a pathway that is safe with regard to triggering TEs.

## 5    Discussion

With CTEM we introduce a simplistic model which is able to represent carbon emissions from the three carbon TEs PFAT, PFTP and AMAZ in line with the estimates from Armstrong McKay et al. (2022). However, the impacts from the carbon TEs estimated in this study remain somewhat speculative, as there is only limited confidence about the actual existence of tipping points within the carbon TEs, with low confidence for PFTP and medium confidence for PFAT and AMAZ (Table 1). The distinction between PFGT, PFAT and PFTP made by Armstrong McKay et al. (2022) for permafrost thaw also remains questionable. Most studies, including the latest IPCC report, assume that permafrost thaw can be divided into gradual and abrupt thaw processes, corresponding to PFGT and PFAT, and do not mention permafrost collapse due to compost bomb instabilities (PFTP) (e.g. Canadell et al., 2021; Turetsky et al., 2020; Schuur et al., 2015; Wang et al., 2023). Since PFTP causes the highest carbon emissions out of all three carbon TEs with up to 250 GtC (Table 1), the impact of tipping elements within the carbon cycle will be less severe than found in this study if PFTP does not include a tipping point.

Yet, additional carbon emissions released by PFGT under continued global warming of up to 260 GtC (Table 1) need to be expected, which are not considered in this study. One reason to exclude PFGT is that it is not assumed to include tipping points, but to be a threshold-free feedback to global warming (Armstrong McKay et al., 2022). Furthermore, sophisticated models of PFGT already exist (e.g. Gasser et al., 2018; Burke et al., 2020), hence it would not be appropriate to represent PFGT with a simple conceptual model like CTEM.

It is challenging to compare the carbon emissions from carbon TEs identified by us to other studies, as this is to the best of our knowledge the first study to investigate the impact of all TEs within the carbon cycle. Other studies are only analysing single TEs, e.g., Cox et al. (2004); Parry et al. (2022) for AMAZ or Schneider von Deimling et al. (2015); Gasser et al. (2018) for permafrost thaw as a whole. Since the individual carbon emissions from the carbon TEs modelled by CTEM are based on the estimates of Armstrong McKay et al. (2022), we would only reproduce their analysis by comparing our individual



carbon emission estimates to the literature. Nevertheless, it must be noted that even though the uncertainty ranges of the carbon

impacts identified by Armstrong McKay et al. (2022) based on numerous studies are large (factor two for PFTP and AMAZ, factor three for PFAT), they do not include all estimates from the literature. Especially, the lower bounds of AMAZ and PFTP are questionable. Carbon emissions from the Yedoma region, which make up for the major part of PFTP emissions, of only 23 GtC in 2300 under RCP5-8.5 have been found by an observations-based modelling study (Schneider von Deimling et al., 2015). This is much lower than the minimum carbon emissions of 100 GtC from the Yedoma region assumed by Armstrong

McKay et al. (2022), which led to an estimated minimum impact of PFTP of 125 GtC. The minimum impact of 30 GtC for AMAZ might also be too high since no substantial carbon loss from AMAZ is observable in CMIP6 models, even if localized dieback occurs (Parry et al., 2022).

A major shortcoming of CTEM is that it does not account for the temporal variability of the $CH_4$ emission fraction of PFAT and PFTP. However, we regard the temporal evolution of this fraction as too poorly constrained to be prescribed in CTEM since

current model studies are rare and vary greatly (compare Turetsky et al. (2020) and Schneider von Deimling et al. (2015)).

Our estimates of probabilities of triggering are generally higher than the values reported by Kriegler et al. (2009), which have found much use in the climate tipping points literature (e.g. Lontzek et al., 2015; Cai et al., 2015, 2016). There, ranges of probabilities of triggering are provided for the year 2200 under three warming scenarios: a low temperature corridor comparable to SSP1-1.9, a medium temperature corridor comparable to SSP2-4.5 and a high temperature corridor comparable to SSP3-7.0.

For the medium temperature corridor, Cai et al. (2016) infer mean probabilities of triggering from the ranges given in Kriegler et al. (2009) of 22% for AMOC, 52% for GRIS, 34% for WAIS, and 48% for AMAZ, which can be compared to our estimates in 2200 under SSP2-4.5 (without carbon TEs impacts) of 38% for AMOC, 94% for GRIS, 98% for WAIS, and 43% for AMAZ (Fig. S15).

Those deviations can be explained by the new scientific literature being published since Kriegler et al. (2009), which pro-

vide the basis for the threshold temperature estimates from Armstrong McKay et al. (2022) that our calculation of triggering probabilities is largely based on. The increased triggering probability for the AMOC can be linked to various studies reporting unrealistic stability of the AMOC in GCMs (e.g. Liu et al., 2017), together with empirical evidence for loss of stability of the AMOC (Boers, 2021). New indications for the proximity of a tipping point for GRIS are also derived from observations of ice loss (King et al., 2020; Boers and Rypdal, 2021), with recent modelling studies confirming this (Van Breedam et al., 2020;

Robinson et al., 2012). Loss of WAIS is becoming observable (Shepherd et al., 2019) and a tipping point might already be crossed with several glaciers in the Amundsen Sea currently undergoing marine ice sheet instability (Rignot et al., 2014).

Even though we find an increase in the probabilities of triggering caused by the additional carbon emissions from carbon TEs, this impact is not strong enough to trigger any tipping cascades. Additional carbon emissions from carbon TEs only cause existing clusters of TEs to be triggered earlier. They do not produce any additional clusters of triggering (Fig. 5). Even

under SSP2-4.5, which features the highest long-term increase in temperature of up to 0.91°C due to carbon emissions from carbon TEs, the additional probability of triggering caused by this temperature increase is only 3.3 pp on average and only two TEs become more likely than not to be triggered if this temperature increase is accounted for. Despite our finding that the





impact from carbon TEs alone is too small to trigger tipping cascades, tipping cascades might still emerge as major physical interactions between TEs aside from carbon emissions are not accounted for in this study (Wunderling et al., 2021).

**6 Conclusions**

Our work indicates that the pathway the world is currently on, which is comparable to SSP2-4.5 (Climate Action Tracker, 2022), is highly unsafe with regard to triggering climate tipping points. There is deep uncertainty about the carbon emissions from carbon TEs under this scenario, but the highest additional long-term temperature increase of all SSPs becomes possible, with the additional warming in 2500 ranging from 0.03°C to 0.91°C (5th to 95th percentile). Temperature impacts from carbon TEs

can be especially high under SSP2-4.5 since maximum carbon emissions from carbon TEs are possible and would constitute a high contribution relative to anthropogenic carbon emissions. Under lower emission SSPs, the risk for high carbon emissions from carbon TEs is significantly reduced, while under higher emission SSPs maximum carbon emissions become more likely but also relatively smaller compared to anthropogenic carbon emissions. Triggering of multiple TEs is likely under SSP2-4.5 with 65% probability of triggering on average over all 16 TEs in 2500. The temperature impacts from carbon TEs increase this

number by 3.3 pp, which is the highest probability increase from carbon TEs out of all SSPs.

The safest pathway with regard to triggering climate tipping points is SSP1-1.9 with an average triggering probability of 28% in 2500. However, even under this scenario the five TEs GRIS, REEF, WAIS, PFAT and BARI become more likely than not to be triggered by 2031. Hence, if the temperature thresholds derived from the literature by Armstrong McKay et al. (2022) are accurate, it may be too late to avoid crossing any climate tipping points.

Since SSP1-1.9 might already be politically infeasible to achieve (Jewell and Cherp, 2020), SSP1-2.6 is potentially a more realistic best-case scenario, associated with limiting global warming to 2°C, which can be achieved if all current climate mitigation pledges are fulfilled on time (Meinshausen et al., 2022). Moving from SSP2-4.5 to SSP1-2.6 reduces the probability of triggering TEs substantially, with an average probability of triggering until 2500 of 38% under SSP1-2.6. The impact of carbon TEs is also less severe under this scenario, ranging from 0°C to 0.53°C in 2500, which leads to an increase in the

average probability of triggering of 2.3 pp.

If the risk of triggering TEs is to be reduced, rapid action is needed to reduce greenhouse gas emissions, since climate tipping points are already close, and it will be decided within the coming decades if they will be crossed or not.

*Code and data availability.* The code to reproduce the model output and the plots is available from
https://github.com/JakobDeutloff/TP_paper and archived under https://doi.org/10.5281/zenodo.8121160 (Deutloff, 2023b). The model output

and the parameter sets used to produce it are archived under https://doi.org/10.5281/zenodo.8099908 (Deutloff, 2023a).



*Author contributions.* JD designed the study and wrote the paper with input from both co-authors. JD performed and analysed the model simulations and prepared all figures and tables. HH co-designed the sampling strategy, and contributed to the discussion of the method and the results. TL contributed to the research design and the discussion of the methods and the results.

*Competing interests.* Tim Lenton is an editor of the special issue on tipping points in the Anthropocene. The peer-review process was guided
by an independent editor, and the authors have also no other competing interests to declare.

*Acknowledgements.* The authors would like to thank David Armstrong McKay and Jesse Abrams for the discussion of ideas and valuable suggestions for the model development.



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
