# Peer review of "High probability of triggering climate tipping points under current policies modestly amplified by Amazon dieback and permafrost thaw"

_EGUsphere, 2023_

## Author Response (AR1)

**Response to the Reviewers**

We thank both reviewers again for the time they dedicated to reviewing our manuscript and the helpful comments they both provided. We have implemented a range of major changes to our document, following your suggestions. Furthermore, we also included most minor changes that were suggested. We list the major changes below, followed by a point-by-point response to both reviewer comments.

**Major Changes to the Manuscript:**

1. We have excluded SSP1-1.9 from our manuscript since it features a major temperature overshoot. Since we are not considering the possibility for tipping elements (TEs) to remain in their stable baseline state if the threshold temperature is exceeded only temporally, we agree that tipping probabilities would be overestimated under SSP1-1.9.
2. We have clarified that CTEM carbon emissions are not included in FaIR in Sec. 2.3.1 and included a section in the supplement to explain this in more detail.
3. We changed the title and reworded the manuscript to better convey our main message, which is that probabilities of triggering TEs are high under current policies but only mildly increased by carbon emissions from abrupt Amazon dieback and permafrost thaw.
4. We corrected a minor flaw in our calculations that slightly lowered probabilities of triggering. In the first version of the manuscript, we forgot to calculate temperature anomalies relative to 1850-1900 before comparing the temperature to the threshold temperatures. This is now implemented.
5. We put Fig. 3 and Fig. 4 on the same y-axis.
6. We don't use the calibrated IPCC language any more.

**Response to Dr Christopher Smith**

How did the authors add CO2 from the carbon-relevant tipping points to FaIR? There are two categories of CO2 in FaIR: fossil & industrial sources (FFI), and AFOLU. Both CO2 sources add CO2 to the atmosphere in the same way, but CO2 AFOLU emissions are used to derive radiative forcing from land surface albedo change (Smith et al. 2018, https://gmd.copernicus.org/articles/11/2273/2018/). This simple relationship is based on the assumption that most CO2 AFOLU emissions are from deforestation for cropland, and changes the land surface from dark to light resulting in a net negative forcing.

At the moment, incorporating emissions from Earth system feedbacks into FaIR is a bit of a hack*. Maybe AMAZ and PFTP would affect forest cover and in both cases, albedo would reduce if the forest dies, so perhaps they should go into the CO2 AFOLU category and PFAT into FFI. It's not FFI of course, but that column can be used for CO2 emissions that do not affect surface albedo and hence no impact on the land surface forcing.

In FaIRv2.0.0 by Leach et al. (2021), which is the version FaIR we are using, albedo shifts due to land use change are prescribed externally and are not coupled to AFOLU carbon emissions. To the best of our knowledge, there is no distinction made between FFI and AFOLU carbon emissions in FaIRv2.0.0, hence there is only one stock of $CO_2$ and $CH_4$ emissions, to which we simply add the emissions from the carbon tipping elements. You are right in assuming that Amazon rainforest dieback would probably lead to a higher surface albedo, but it is also predicted to reduce cloud cover and this is the greater effect. The total biogeophysical effect of tropical forest loss is estimated to be positive, i.e. forest loss would warm the earth beyond the warming induced by the emitted carbon alone (Lawrence et al. 2022, https://www.frontiersin.org/articles/10.3389/ffgc.2022.756115/full ). There are several mechanisms at play producing this positive feedback, with the main ones being reduced surface cooling by evapotranspiration from forests, which also leads to reduced cloud formation. Permafrost thaw is assumed to promote northward migration of boreal forests, which would decrease the local albedo (Scheffer et al. 2021, https://www.pnas.org/doi/10.1073/pnas.1219844110). However, this albedo change is more related to the northern expansion of boreal forests, which is also assumed to include a tipping point. We do not consider changes in boreal forest cover, since northern expansion and southern dieback are expected to roughly balance out.

Including biogeophysical effects in our study would require a significant increase in complexity, without necessarily increasing its meaningfulness. Hence, we decided to disregard them and focus on the effects of carbon emissions alone.

[as an aside I don't find some of the acronyms for the tipping points very intuitive. I'd almost recommend spelling them out everywhere, if you can stomach it].

We decided to spell them out in the introduction, the abstract, and the conclusion, but to abbreviate them in the methods and results section. For once, this saves some space, but more importantly it keeps the text comparable to the figures, where we have to abbreviate.

Starting from section 3: I'm uncomfortable about applying IPCC calibrated language to the findings made in this paper, since the IPCC statements relate to assessments made with multiple lines of evidence than apply nominal probability ranges to likelihood statements, and this paper presents one study with calculated probabilities of tipping points being breached. Using a different calibration of FaIR would give you different results, as would using different probability distributions for your tipping point threshold crossings (see two comments below on these points). The discomfort comes when translating probabilities from this study back into natural language. One example is given on line 327: "(GRIS, WAIS, REEF and PFAT) become more likely than not to be triggered by 2026". Is this really true? This seems like a very over-confident statement to make with a very precise timeframe given. In short, the relationship between IPCC calibrated language and probabilities flows in one direction, but not both.

Thanks for the remark. After spending a lot of time to come up with a reason to keep it in our initial response to you, we decided to not use the IPCC calibrated language any more since the second reviewer and the editor also recommended that.

One thing I am missing a little from this study is the consideration of overshoot. My reading is that once a temperature threshold is crossed, a tipping point will happen with certainty,

though it may take hundreds of years to manifest and in the meantime we may have been able to bring temperatures down substantially. Can we tolerate overshooting a threshold, if the overshoot is small and temporary? Are any of the tipping points reversible on the lower branch of the bifurcation if temperatures are reduced?

This is actually a major shortcoming of our study, therefore we decided to remove SSP1-1.9 from our analysis since it features a pronounced temperature overshoot. Since there is only a mild overshoot for SSP1-2.6 we keep it but mention the effect in the discussion section. Including internal timescales in our analysis is not an option, since they remain largely unconstrained.

9: FaIR is not an intermediate complexity model in the sense of it including a gridded, book-keeping land and ocean carbon cycle module such as in a model like UVic. It's much simpler; "emulator" or "reduced complexity model" is more appropriate.

We changed that, thanks.

13: "triggering until the year 2500 of 65%". I'm not sure this statement is well-defined. Triggering which tipping point? All of them?

This refers to the probability of tipping on average over all tipping elements in the year 2500. We have changed the sentence and hope the point is more clear now.

60: under what process do the NH and SH forest dieback balance out in terms of global warming? Is it a carbon cycle feedback, surface albedo feedback, some combination of both?

It actually depends on whether you believe the IPCC or Armstrong McKay et al. (2022). We included a section on that in the introduction.

64-65: difficult to read sentence: would recommend separating list items with semicolons.

Implemented, thanks.

66: FaIR does not include the deepening of the active layer / gradual permafrost release feedback either, so I assume that this process is not accounted for in your analysis.

Correct, we do not include gradual thaw of permafrost since it is not regarded to be a tipping element (Armstrong McKay et al. 2022) and more sophisticated modelling techniques already exist.

71: 20% - is this the fraction of SOC released as methane? Please confirm.

Yes, we stated this more clearly now.

115: best to specify FaIR v2.0.0. Since there is a v2.1 already out. In fact the calibration of the model makes rather a large difference (lines 121-123), as we have a constrained ensemble of FaIR v2.1 which meets all of the IPCC assessed constraints with good precision as well as historical climate observations (https://zenodo.org/record/7694879).

Thanks for the hint, we did that.

Table 1: while a review of Armstrong McKay et al. (2022) and this paper does not provide a new analysis of tipping points, I would think that Sahel greening would constitute some carbon drawdown.

Armstrong McKay et al. (2022) mention in the supplementary material that some carbon would be sequestered by Sahel greening, but not enough to measurably impact the global climate and land surface albedo would be lowered.

202-210: the list of distributions to choose from is limited by the software package, which is unfortunate. Any three given percentiles of a distribution can be fit with a three parameter model; a good choice here would be skew-normal, which reduces to a normal if the upper and lower bounds are symmetric.

Thanks for pointing that out. Using a three parameter model would probably improve our methodology, however we are not in a position to redo the whole analysis. Furthermore, the distributions we use are fitting the percentiles sufficiently good.

Figure 3, 4 (maybe others): using the same y-axis range on each subplot would be more informative.

We followed your suggestion, thanks.

289: we don't take credit or blame for the forcing relationship in FaIR; it is derived from Etminan et al. (2016) at https://agupubs.onlinelibrary.wiley.com/doi/full/10.1002/2016GL071930, which is a fit to line-by-line radiative transfer simulations.

Changed accordingly.

Table 2: SSP1-19 results have bigger dT in 2300 than 2200 and 2400; what's going on here?

This can be linked to declining atmospheric methane concentration anomalies due to the comparably short lifetime of methane. We mention this in line 302.

311: The chance of TEs being triggered earlier if carbon emissions are included is surely 100%? Because a carbon TE doesn't ever remove carbon, and your delta warming will always be positive unless your additional carbon release is zero.

That is correct, we have changed this sentence to "Especially TEs within the cryosphere will become more likely than not to be triggered decades earlier, if carbon emissions from carbon TEs are included."

315: "disproportionately": not sure about use of term here

We have changed it to "significantly".

**Response to Reviewer 2**

The manuscript by Deutloff et al combines assessments of thresholds of tipping elements in the earth system based on a recent paper by Armstrong McKay (expressed in GMST) with the uncertainty of different temperature outcomes under SSP scenarios. This is a relevant and useful idea, as the focus on GMST levels alone misses out on the scenario uncertainty with regard to different warming outcomes. However, it is also not entirely novel as e.g. Kloenne et al (2022) make similar points for irreversible thresholds in the cryosphere.

However, there are several substantial shortcomings with the analysis in its current form. Some of the results about timing of tipping points I'm not sure are correct. But more importantly, the way tipping is implemented without any considerations of the temporal dynamics of GSMT is too simplistic.

Also, the deterministic nature in which values of an expert assessment in a single study (which is not IPCC) are being implemented requires some reflection.

We make it clear that our study depends on the findings of Armstrong McKay et al. (2022) in paragraph 2 of the introduction. It is our understanding that this study is currently the best summary of the current literature on tipping points, hence it represents a reasonable basis for our analysis. We made this point more clear in the last paragraph of the discussion.

The manuscript has quite some focus on tipping elements with carbon cycle relevance and attempts a coupling between their carbon cycle model and FAIR. I'm not convinced this analysis is robust – first and foremost because the carbon cycle in FAIR is not well constrained and assuming that these dynamics would really be 'additional' is an assumption that needs to be justified. However, no analysis of the carbon cycle dynamics in FAIR is presented. I generally think this might not be the right SCM to attempt such a coupling.

We included a two sentences in Sec. 2.3.1 to mention explicitly why we think FaIR is suited for such an attempt. We give more detail on that in Sec. S6.

Secondly, the results are being highlighted quite a bit across the manuscript – whereas in my mind the real finding is that their effects are very small. However, even the title of the manuscript seems to imply that there's something more to it. I understand that this might have been the hypothesis of the authors when they started their analysis. But their results don't show it (there are also some issues with how the additional GMST is derived if I understood it correctly). I'd suggest the authors may want to consider removing this part of the analysis, or else substantially repackaging it and underscoring the explorative and illustrative nature of the analysis presented.

We agree that the overall effect of carbon emissions from carbon tipping elements is

small compared to anthropogenic emissions. We made this point more prominent in the results and the conclusions, and included it in the title.

L 61: This strikes me as an argument that's somewhat incoherent with the rest of the manuscript that (rightly) focusses on uncertainties. Because even if they may cancel out in the central estimate, any deviations from this could still have potentially far reaching implications. Also note my comment on the numbers in Table 1.

It is correct that adding BORF and TUND to our analysis would lead to a larger uncertainty of carbon tipping elements impacts, even if we assume their carbon emissions cancel out, which is not the case in Armstrong McKay et al. However, we think it is reasonable to exclude BORF and TUND since they are not expected to be major tipping elements within the Earth's carbon cycle according to the IPCC and their biogeophysical effects (mainly albedo) predominate the impact their carbon emissions would have on global warming. We made this point more clear in the introduction (paragraph starting from L 61).

L89: This assertion is not correct. A delayed action (past 2030) 1.8°C in 2100 scenario as identified in Meinshausen et al. is quite different from SSP1-2.6 (with emission reductions starting in 2020) – also in the long run which matter a lot for the outcomes here.

You are right, the two scenarios are not exactly identical. We omitted this comparison from the manuscript.

L89: I don't think this paper is the place to speculate what's political feasible or not. SSP1-1.9 is part of the core set of IPCC WG1 and the WG3 has published a range of IMP scenarios that resemble similar characteristics (1.5 low and no OS). The authors are of course free to choose whichever scenario they like – but to argue SSP1-1.9 was not but SSP1-2.6 is not convincing. Also, it's long-term outcome emissions and temperature trajectory that matters more than the next decade for the tipping dynamics here I understand. And what's "feasible" on these scales is not established.

Point taken, we removed this bit as it is no longer necessary with SSP1-1.9 removed from our analysis.

Table 1: I'm a bit surprised by the numbers for BORF and TUND. The carbon removal is an order of magnitude different, but the warming effect is very similar. I understand that's also the case in the original Armstrong-McKay paper. But would be good to verify and explain these findings. I am not across the underlying literature on this – but if the reason was biophysical effects (i.e. warming by increased tree cover), then this would be a local effect rather different from the one on the global carbon cycle.

We explain this in more detail in our introduction now (paragraph starting from L 61).

L 138: I would like to get some more clarity on why this assumption that PFAT is amplifying PFGT is justified.

Armstrong McKay et al. (2022) base this assumption on the finding from Turetsky et al. (2020) that PFAT spreads at similar rates as PFGT in permafrost models. We included this point in the introduction (L 81).

L 175: I'm a bit concerned about this implementation as there's substantial variability in the carbon cycle response across the full FAIR member ensemble. Some of these carbon cycle ensemble members may actually already reflect (at least conceptually) some high emission outcomes including from those sources assumed (although they're of course not explicitly modelled in FAIR). So right now it appears to me that some additional emissions are added to ensemble representations that may already, at least by allowing for a wide uncertainty range during constraining, account for some of the effects considered. In other words, I find it very hard to argue that these modelled TE effects are really "additional" when considering the wide range of carbon cycle outcomes under FAIR.

So I'm not sure this approach actually works – or is a bit overly simplistic. Some other simple climate models such as OSCAR have a much more detailed representation of the carbon cycle including also a permafrost module, for example. They might be much better suited for such an application. Else it might be better to remove that part of the analysis.

We are aware that the possibility for double-counting of carbon emissions exists, and therefore checked in detail whether this is the case. We did not include a section on this in our first version of the manuscript but added a sentence which makes this point clear in the revised version (L 199). Furthermore, we added section S6 to the supplement, which explains this in more detail.

L195: So to make sure I get this right: Distributions are fitted through 3 data points based on expert assessment, is that correct? It seems to me that pre-industrial = zero risk is also fixed, right? I think it's fair to say that these distributions are then not very well constrained, also bc. the assumption of taking values for min/max/ best estimate as a given without assuming (allowing for) uncertainties around them. It would be good to see some sensitivity studies of fitting different distributions with different rigidity to assess the effect.

Your interpretation of the probability distributions is correct. We did not include uncertainties of the min/max/best estimate because this range of estimates is already intended to represent the uncertainty in it. Hence, it would not be useful to include an "uncertainty of the uncertainty" by guessing uncertainty ranges of the min/max/best estimate. We did not add any sensitivity studies, because, after serious consideration, we don't see the additional value of this. The uncertainty in any of the estimates is contained in the range of the min, max values given by Armstrong McKay et al. (2022) and it would not make sense to randomly vary them as part of a sensitivity study. Concerning the fitted distributions, we simply fitted them as good as we could. While it might make sense to use more complex distributions to better fit the estimates as suggested by Dr Smith, we don't see the additional value in varying them (which would necessary mean making them fit less good) as part of a sensitivity study.

L230: This part strikes me as crucial and I don't know if agree with the approach taken here. It is my understanding that the assessment made in Armstrong-McKay relate to stabilization temperatures. But it is not well established for how long these temperature levels would need to be exceeded in order to trigger tipping. If I understand the proposed methodology correctly, this would not be taken into account. If peak warming is above a randomly sampled value, it's triggered – regardless of the temperature trajectory after. I don't think that works. As in particular for some of the elements considered, i.e. sea-ice, they would respond quite quickly to a reversal of global temperatures. Similarly, the AMOC for example might show a rapid recovery or even overshoot under reversal of warming (at least in relation to its thermal component – the saline component would probably need to consider a coupling to the Greenland ice sheet). I'd argue that this would also matter for the permafrost dynamics quite a bit, in particular PFAT – that should be stopped once temperatures decline below again

Other approaches such as by Wunderling et al (2022) explicitly take this time dimension into account and show that long-term stabilization temperatures actually matter quite a bit. So with this current implementation, tipping risks under SSP1-2.6 and SSP1-1.9 are systematically overestimated. As this is also quite apparent in the results (i.e. Fig. 4) I think this should be addressed. I also think it shouldn't be all too difficult to come up with a temporal distribution for "overshoot" time coupled to peak warming and test the sensitivies of the outcomes towards considering this effect.

We agree that this is an important point and have therefore removed SSP1-1.9 from our analysis, since this scenario includes a pronounced temperature overshoot. SSP1-2.6 still includes a mild temperature overshoot, but we think it is sufficiently small to include it. We mention the potential for overestimating risks of triggering under SSP1-2.6 in the discussion.

Since the cumulative carbon emissions from PFAT scale linearly with the surface temperature anomaly (eq. 2), the missing internal timescale is not a problem in this case.

To include internal timescales in our analysis would increase the complexity of our approach beyond the scope of this study. Furthermore, we don't think they are sufficiently constrained yet to do so.  However, it could be an interesting opportunity for future projects building on our work.

L245: I suggest to not use IPCC calibrated language here (but rather stick to the percentiles). This study is explorative and in this way interesting, but still very far away from the robustness in understanding that would underly any IPCC assessment.

Agreed, we followed your suggestion.

Figure 3: Strongly suggest to put them all on the same y-axis. (Or at least group together). This way the first visual impression of what this graph is saying is quite misleading.

Thanks, we did that.

L275: This comparison to the median of the ensemble doesn't make much sense. Clearly, the high end TE feedback outcomes, would be triggered under high warming FAIR realisations. So they wouldn't materialize compared to the median and their relative contribution would be smaller. I suggest to derive the additional warming relative to each individual realization.

You are right, high-end TE feedback outcomes are more likely to be triggered under high warming FaIR realisations, this is why the 95th percentile rise more than the median of the temperature distributions as shown in Figure 4. We included such a sentence in Sec. 4. However, in Table 2 we show exactly what you propose deriving. Here we calculate the additional warming we get for every individual ensemble member from including carbon TEs and then calculate the percentiles from this additional warming. We hope the caption of Table 2 is more understandable now. We also rephrased this section quite a bit and stated the main message more clearly at the end.

Table 2: Why is there a peak in 2300 despite methane and $CO_2$ emissions staying pretty constant for SSP2-45 and lower scenarios? Is this only because of the PFAT component? I find this a bit strange tbh. And would suggest the authors look into this more to understand what drives this behaviour (might well be an artefact of their method to derive warming relative to the median, also noting that the uncertainty ranges don't change as much as the median).

The methane and CO2 emissions in Table 2 are cumulative, so after 2300 there are no major additional emissions. The temperature decrease after 2300 can be explained by the declining atmospheric methane concentrations (Fig. S10).

L300: Not sure I understand what is meant here. Methane concentrations should decrease even faster without those additional emissions. So any additional source should keep the warming up implied by the rate of emissions pretty much. Maybe the authors can help me out here.

Emissions in Table 2 are cumulative, and the cumulative carbon emissions from carbon TEs stop increasing towards the end of the model period. Given the comparably short lifetime of methane, additional atmospheric methane concentrations caused by tipping of the carbon TEs decrease towards the end of the model period, and so does the additional warming. We show the decrease in atmospheric methane concentrations in Fig. S9.

Fig. 5: This figure illustrates the problems with this approach. Absence of a temporal component makes all tipping elements leaves almost no scenario dependence in the near-term, but the signal is determined by the median warming trajectory. It then also seems to imply that 5 tipping points are breached in 2025 under all scenarios. I'm not convinced this actually represents dynamics of the systems under investigation and that the evidence for such an imminent tipping is sufficient. I'm also a bit confused timing-wise. The threshold for GRIS for example is established as 1.5°C (median estimate) – but the crossing time here is 2023 or 2025. Similar for REEF and WAIS, as well as PFAT. That's more around 1.3°C and 10 years earlier than when 1.5°C would be

crossed in the SSPs in FAIR. I'm not even sure if 1.5°C is exceeded in SSP1-1.9 in FAIR (certainly not by much and for long). I appreciate that there's some skewness introduced by the fitted distributions, but by eye-inspection this doesn't look like so much from Fig. 2. So I suspect there's actually a mistake here – which would need to be corrected.

Thank you for raising this point, as mentioned, there was actually a mistake here. In our code, we forgot to calculate the temperature anomaly relative to the 1850-1900 period before comparing it to the tipping thresholds but used the raw temperature. The raw temperatures produced by FaIR actually cross 1.5°C in the median under SSP1-2.6 in 2025, which explains that our model produces a 50% chance of tipping for TEs with a best estimate of 1.5°C in this year. The temperature anomaly relative to 1850-1900 only crosses 1.5°C in 2027 in the median, so this error leads to crossing of the 50th percentile two years earlier. We have corrected this and updated all numbers accordingly.

L367: Small compared to what? And the fact that there's no scenario dependency in the timing of some of the tipping points is a direct outcome of your assumptions including of not considering temporal dynamics from tipping (and maybe some errors in the GMST estimates from FAIR?).

We removed this sentence since SSP1-1.9 is no longer included.

L378: Agreed re questionable assumption on permafrost. Maybe a good reason to not do it?

Since we use Armstrong McKay et al. (2022) as the foundation of our study, we think it is reasonable to follow their suggestion on how to divide between permafrost components. However, our analysis of additional carbon emissions from carbon TEs must to some degree be seen as hypothetical, given the partly low confidence in their existence. This is the message we want to convey with this paragraph. We hope this becomes more clear now.

L445: See comment above on SSP1-1.9 and feasibility discussion. Please revise

SSP1-1.9 was removed.

---

## Author Response (AR2)

**Response to the Reviewers**

We thank both reviewers for the time they dedicated to reviewing our manuscript a second time and the helpful comments they both provided. This is a great help for improving our manuscript and highly appreciated. Since both the second reviewer and the editor asked for major revisions, we followed their suggestion, hopefully this time satisfyingly. We list the major changes below, followed by a point-by-point response to both reviewer comments.

**Major Changes to the Manuscript:**

1. To show that our approach of assuming a tipping element (TE) is triggered instantaneously once its threshold temperature is crossed leads to an overestimation of probabilities of triggering for scenarios including a temperature overshoot, we introduced the concept of delayed triggering. We added a section to the introduction explaining the concepts of instantaneous vs delayed triggering and why we need them, and adapted our analysis accordingly. The major changes that come with this are:

   a. We adapted the burning ember plots by removing the colour gradient, which was not adding any information, and instead included a third bar that represents the probability of delayed triggering.
   b. We included our new results in section 5.
   c. We added a section to the discussion in which we elaborate on both concepts and why this shows that we need to better understand the timing aspect of TEs.
   d. We added our main finding from this part of the analysis to the abstract and the conclusion.
2. Due to the warm bias of FaIRv2.0.0 and the difficulty of interpreting our results, we removed the first part of section 5 and figure 5 that were focussing on the timing of triggering TEs.
3. We include SSP1-1.9 again to show how uncertain probabilities of triggering are in the case of temperature overshoot, due to our limited knowledge about the effective timescales of TEs.

**Response to Dr Christopher Smith**

Thanks to the authors for taking the comments on board from the first revision. This is now a better paper. In my opinion there are still a few things to tie up.

We thank you for your second round of helpful comments. Since the second reviewer and the editor asked for major revisions, we have changed the manuscript quite substantially. We hope this will make it an interesting read that can convince you to participate in the third round of revisions.

Abstract line 13: "Averaged over all 16 tipping points, the probability of triggering until the year 2500 is 64% under SSP2-4.5". I'm still struggling to parse this statement. 64% of triggering one of them? all of them? Eight of them?

What we mean to express with this is that when we have probabilities of triggering until 2500 for all 16 TEs in percent and average over the over those 16 values, the mean is 64%. It is the same value reported in Table 3. We have changed the structure of the sentence slightly and hope it becomes more clear now.

Line 44-45: "regional impact TEs are required to either contribute significantly to human welfare or to have great value in themselves as unique features of the Earth system". "Contribute significantly" and "have great value" makes it sound like TEs are a good thing. It is contradicted (presumably correctly) at the start of the next paragraph. I'm sorry I didn't remark on this first time.

True, that might be a bit confusing. It should say that in their intact (not tipped) state regional TEs are valuable to the society, hence tipping is dangerous. We included this distinction.

Line 90: Please explicitly state FaIRv2.0.0 here.

Done

Line 95: pedants' corner. SSP1-1.9 is a Tier 2 scenario technically in O'Neill et al. and CMIP6. So you could just delete "except for SSP1-1.9 (O'Neill et al., 2016)".

Thanks for the hint, we must have missed that. We now speak of five SSPs and leave out the statement about the Tier x.

Line 113: "five different SSP scenarios". Now four.

Now back to five again.

Line 124: "Effective radiative forcing for CMIP6 follows the data provided by Smith (2020)." The previous few sentences describe all the sources of emissions to run FaIR with (you could also simply cut these and just cite Lewis & Nicholls 2021, https://zenodo.org/records/4589756 and Nicholls et al. 2020, describing the preparation of the emissions for RCMIP), so you shouldn't need to use an external forcing time series. Perhaps you did use the external forcings for land use, solar and volcanic, in which case state this.

We state your last point more clearly now, sorry to not have included this earlier. However, we decided to keep the references to all emission datasets, since we mention the scenario extensions from Meinshausen et al. (2020) in our discussion.

Lines 132-135: I have to state again that the calibration of FaIR (or any other reduced complexity climate model used) makes a huge difference to the results. In the Leach paper, of which I was a co-author, we didn't constrain FaIR v2.0.0 to the same level of rigour as

FaIR v1.6 was for the IPCC AR6, or v2.1 is currently, where the IPCC constraints (on climate sensitivity, ocean heat content, aerosol forcing and future warming in SSP scenarios) ensure the projections from the model follow the best available science. Using one of these calibrations would give you a 95% ECS at or very close to 5°C.

For example, SSP1-2.6 should not be crossing 1.5°C in 2027 in the median, which was mentioned in your earlier response (IPCC WG1 Chapter 4 has all SSPs crossing 1.5°C in the early 2030s). It's likely that the Leach calibration runs warm. Therefore you may see earlier triggering of TEs in this study than I would intuitively (though I'm not a TE domain expert) expect.

Sorry for labouring the point, but getting projections out of FaIR that are suitable for climate policy recommendations is basically my day job. I just want to see something in this paragraph along the lines of "using the calibration of FaIR in Leach et al. (2021)".

You are right, we have made this more explicit to stay fair. The point that v2.0.0 is not AR6 calibrated while v1.6 and v2.1 are is actually something we wish we had realised before designing our model experiment. We checked whether it would be possible for us to switch to any of those versions for our revision. However, the implementation of FaIR seems to be quite different between the different versions, which unfortunately makes this endeavour infeasible for us due to time constraints. To not make overexaggerated statements about the timing of triggering TEs, we have removed the time dimension from this part of our analysis, which is also something the second reviewer requested.

Line 300: SSP3-70 -> SSP3-7.0

Done, thanks.

Line 318: "where triggering becomes more likely than not": suggest to delete this, since it is indicative of IPCC calibrated language and adds no additional information to the previous statement on 50% probability. Appreciate this wasn't necessarily the intention. Please also change "becomes more likely than not to be triggered" to "is triggered with greater than 50% probability" in line 341.

That's another valid point, however, we have removed this part of our analysis so it doesn't apply any more.

**Response to Reviewer 2**

I want to thank the authors for their efforts in addressing my comments and the comments of the other referee. I also want to apologize for the delay in providing my assessment due to personal reasons.

We want to thank you for your detailed comments, which show you have really engaged with our study. We hope that our implementation of your feedback meets your expectations this time and helps to improve the quality of our work.

But I have to say, I am a bit disappointed by the authors response to my comments. There are some very fundamental issues in relation to their assessment of tipping element timing and outcomes. And rather than addressing the issue, the authors have chosen to delete the scenario (SSP1-1.9) where these issues are most apparent. That doesn't solve the problem and I don't agree with that approach. In fact, I think it would be important to keep SSP1-1.9 in to illustrate the point.

To be more clear what the issue is: The manuscript deploys temperature thresholds for tipping dynamics that assume constant temperature levels over millennia. That's fine, that's all we have so far. But let's call them long term tipping points (LTTPs)

But this does obviously not imply that the tipping will be initiated as soon as this LTTP is exceeded for the first time. In fact, there's very little that can be said about when exactly, time-wise tipping will be initiated. Certainly not with decadal or even sub-decadal resolution.

In response to my earlier suggestion, the authors replied that "To include internal timescales in our analysis would increase the complexity of our approach beyond the scope of this study. "

That's fine by me. But this in turn means that all results implying a timing of tipping need to be removed. Take i.e. Fig. 5 and subsequent figures. What they show is not when "tipping risks" are crossed. But when the temperature level of a multi-millennial LTTP is exceeded for the first time and thus a 'zone of elevated tipping risks. The authors may want to think about an appropriate framing here.

This either needs to be substantially revised to make that clear, or the temporal dimension needs to be removed. It might be cleaner to do the latter, because interpreting the former is difficult.

The authors may argue that the timescale until 2500, elevated temperatures above that level would lead to tipping with a sufficiently high chance considering the internal dynamics of the system. I'd like to see this discussed and substantiated but would agree that this is straight forward for all tipping elements but ice sheets, for which one need to argue this carefully. So this main part of the analysis, what tipping risks are until 2500, could be sustained,

The inclusion of the Amazon and Permafrost feedbacks are a bit of a problem, too, but if you assume this happens by 2300 at the latest you might be fine if I interpret your setup correctly.

Taking such an approach also illustrates why it's actually interesting to have SSP1-1.9 in the set, rather than out. Because one could show that the tipping risk under this scenario would be substantially, if immediate tipping was assumed at peak warming. Or much more moderate, if the long-term warming outcome was considered. And since you can't say which one of the two is true, you'd need to present them both, which I think is important in terms of directing future research and to inform your conclusions.

Thank you for this clear description of the problem. We followed your suggestion to include SSP1-1.9 again and removed the timing aspect from our analysis of probabilities of

triggering. Furthermore, we introduced the concept probabilities of delayed triggering and made or assumptions about the timescale more explicit by calling our previous probability estimates probabilities of instantaneous triggering. The difference between the two shows how the unknown effective timescale of TEs keeps us from making more precise statements about the probability of triggering in the case of a temperature overshoot.

It would also illustrate that there's an issue using this long pathway extensions for face value. They are useful tools for exploration, but not a 'given'. There's no IAM scenarios underlying those or else. And societies may well choose to bring temperatures down again between 2100-2500 if tipping risks loom large. This needs to be discussed.

We agree, the probabilities of delayed triggering only rely on the stabilized temperature and hence strongly depend on the scenario. We include this point in L437 — L440.

Other comments:

The Carbon cycle additionality question.
The additional explanations in the SI are helpful. Thank you. And I tend to agree for permafrost, because this is a dedicated process that's either there or not. For Amazon collapse, I find it to be a bit more complicated. Because it's not clear to me to what extent an Amazon collapse by and in itself is actually additional or different with regards to a scenario where the forest stays, but it's role as a carbon sink is greatly diminished. I'm intrigued by the authors statement that GFDL-ESM4 predicts abrupt dieback in parts of AMAZ in the 1pctCO2 run, but that this does not lead to significant reduction of the total carbon being stored in the Amazon vegetation. It's not my area of expertise and I haven't looked into this further. But it seems to substantiate this concern of mine that arguing this to be fully additional is a tricky.

We are glad that the additional explanations have proven to be helpful. Regarding the Amazon, we are still convinced that it is valid to assume carbon emissions from it as modelled by CTEM can be seen as fully additional to the FaIR carbon cycle with no double counting involved. The way we and Armstrong McKay et al. (2022) define carbon emissions from Amazon collapse only includes carbon that would be emitted from the degradation of the forest. The estimate is based on scaling up carbon storage estimates of the two vegetation types forest and savanna (SI of Armstrong McKay et al. (2022)). We make this more clear in L89 now.  Any scenario in which the forest stays and is not converted to steppe, even if its role as a carbon sink is greatly diminished, does by definition not include Amazon collapse. Of the 11 ESMs used for the calibration of FaIRv2.0.0, only three would be able to represent such a change by including a dynamic vegetation model. However, even GFDL-ESM4, which predicts local dieback, does not include wide-scale conversion of forest to savanna. Therefore, we conclude that Amazon collapse is not included in any of the models used to parameterize FaIR and can hence not be included implicitly in the FaIR carbon cycle.

FaIR temperatures.
There seems to be something off with the ensemble the authors are using. In fact, they acknowledge this even in L130-135.

An AR6 constrained FaIR would not reach 1.5°C in 2025 in SSP1-2.6. That's quite outside the range. In fact, FaIR 1.6.4 only reached 1.5°C in current policy in 2040… And the AR6 assessment is sometime between 2030-2040. So the Fair 2.0 Leach et al. configuration deployed seems not ideal and the authors should be advised to revert it to one that's AR6 calibrated – or else provide a comparison that would increase transparency. What doesn't work, of course.

This also implies that there's something off here that would imply higher than AR6 peak warming for the same scenarios. Suggestion is to check 2010-19 warming and compare with AR6 WG1.

You are right, FaIR is warming too quickly in the early 21st century. This is due to version 2.0.0 of FaIR not being fully AR6 calibrated, as was brought to our attention by Dr Smith, the second reviewer and one of the developers of FaIR. We considered switching to FaIRv1.6 or FaIRv2.1, which are both AR6 calibrated, however, this has proven to be technically infeasible since the implementation of both versions is entirely different from the one we are using. Hence, this would mean starting from scratch, which we don't have the time for. However, with the timing aspect removed from our analysis, we think that this point is not as important any more. We acknowledge that the climate sensitivity of FaIR is not well constrained towards its upper end and included the implications in the discussion (L433 — L436).

The question of fitting the distribution:
I appreciate that you fitted those 'as good as you could'. To parameters which are not well constrained at all. I.e. are the authors certain that the max for a given threshold is really 5°. And not 4.9°C or 5.1°C, etc. So I think good arguments can be made to relax those criteria, by how much would your distributions widen? And, given that warming outcomes are skewed high, as are the LTTP distributions, how much does it matter? I don't argue it's critical that this is done, but it should be acknowledged in my view.

We agree that the percentiles given by Armstrong McKay et al. (2022) are not well constrained and make this more clear now in L247. We also agree that this would allow us to relax those criteria and play around with different probability distributions. However, we don't see the additional value of discussing this in our manuscript without actually doing it. Since we don't think that our results would change much if we alter the probability distributions while staying somewhat close to the estimates from Armstrong McKay et al. (2022), which are the only reference point we have, we hope the referee is OK with us not including this point in our analysis. Altering the probability distributions of $Q$ would in our opinion only make sense if we had better information about what they could be, which requires further research to be conducted. We acknowledge this point in the last paragraph of the discussion.

---

## Author Response (AR3)

**Response to Reviewers**

We thank both reviewers again for the time they dedicated to reviewing our manuscript and the helpful comments they both provided. We have changed the manuscript following the suggestions of reviewer one. Please find our point-by-point response below.

**Response to Reviewer 1**

M1 - Timescales: The authors state that the (effective) timescales are set to zero (L 99), and this shortcoming is discussed in several parts of the manuscript. While I definitely appreciate that, I am wondering why the authors still state tipping probabilities in the year 2500, and I don't think their current rationale holds (ice sheets have longer timescales, also considering their potential to trigger further subsequent tipping elements). Also, the authors cannot (and also do not claim) to make a difference between triggering and realization of the tipping element.

Given these shortcomings and in line with reviewer 2, I recommend to get rid of the timescale notion (in particular the 2500) where timescales are actually not taken into account. Rather, the authors could speak of crossing tipping points for equilibrium temperatures (that are different for each of the SSPs) because that is what the authors essentially do.

As such, I also think that the difference between tipping probabilities between (i) Delayed, (ii) Instantaneous tipping and (iii) Instantaneous tipping + carbon feedbacks is a very neat and helpful differentiation and analysis. Still, since the authors cannot take into account the timescales of tipping, (i) could be seen as a lower bound and (ii) as an upper bound of tipping probabilities – this could even be framed like this in the paper (if the authors agree). The current framing as instantaneous and delayed tipping, however, again doesn't really make sense because it gives the impression that timescales are investigated but they are not. Therefore, a different wording is needed.

Summarized, I think the many places in the manuscript that still indicate these timescales should be rewritten (in particular abstract, conclusion/discussion, and figure captions).

Thank you for reconsidering this point. We agree that concentrating on the 2500 time horizon might be confusing and doesn't add much additional information. Therefore, we do not use this notion any more, but speak of the probabilities in a more general sense as probabilities of triggering given a certain SSP.

We also renamed the "probability of delayed triggering" to "probability of equilibrium triggering", recognising the point the reviewer raised about this being the response to equilibrium temperatures. However, we still think that it makes sense to define instantaneous and equilibrium triggering by the different assumption we make about the effective timescale

of the tipping elements. This allows for a more meaningful discussion of the source of the uncertainty expressed by the difference of the two estimates, namely the unknown effective timescale. We mention that the two estimates can be interpreted as a lower/upper bound of triggering probabilities in L 109, if TE interactions are not considered. Adding the carbon TEs shows that the upper bound of triggering probabilities (instantaneous triggering) is increased by TE interactions.

The different probability estimates are now introduced as follows (L97-L111):
*Instead, we adopt the most simple case for which the effective timescales are zero, i.e., a TE is triggered instantaneously once its threshold temperature is crossed. To show that this approach would lead to an overestimation of probabilities of triggering for emission scenarios producing a temperature overshoot, we also discuss the case of equilibrium triggering. Here, we assume that the effective timescales of the TEs are long compared to the overshoot time, i.e., a TE is only triggered if the stabilized temperature at the end of the model period exceeds the threshold temperature. The real probability of triggering will be somewhere between the probability of instantaneous triggering and the probability of equilibrium triggering, but remains unknown.*

*To analyse how carbon TEs and our assumption about the effective timescale of TEs affect the probabilities of triggering, we derive three estimates of the probability of triggering with different degrees of conservatism: equilibrium triggering, instantaneous triggering, and instantaneous triggering including the effect of carbon TEs. Distinguishing between the probabilities of instantaneous and equilibrium triggering allows us to estimate the magnitude of the uncertainty in the triggering probability resulting from not knowing the effective timescale of the TEs. The probability of instantaneous triggering can be interpreted as an upper bound and the probability of equilibrium triggering as a lower bound on the probability of triggering if interactions between TEs are ignored. The third probability estimate allows us to investigate how much the upper bound of the triggering probabilities could be increased by carbon TEs.*

We explain in more depth how those probability estimates are derived in the methods section (L264 - L270) and discuss the implications of the different estimates in L422-L429.

M2 – calibration: L 112-121: I understand that the authors chose the Leach et al. (2021) calibration of FaIR, which is unfortunately not calibrated to the IPCC AR6 runs. While this is a shortcoming, I understand that this cannot be changed anymore at this point in time. That's fine with me, but I think a clear statement (and maybe a brief limitation discussion, how large are differences to be expected) of this in the methods is required.

We agree this needs to be made clearer. We now discuss this point more thoroughly, starting from L 430:
*The probabilities of triggering derived by us might be slightly overestimated, since by the climate sensitivity of FaIRv2.0.0 is not well constrained towards its upper limit (Leach et al., 2021). This is a result of this version of FaIR not being calibrated to match the IPCC range of climate sensitivity. This has been fixed in later versions of the model we were not aware of when conducting this study. Nevertheless, we regard this possible overestimation to be*

*small, since the median climate sensitivity of Fairv2.0.0 agrees well with the latest IPCC estimate (Forster et al., 2021).*

Minor comments:

1) The authors find that the tipping effects from the feedbacks of the carbon tipping elements are relatively minor (around 3% tipping probability difference). That is a nice finding and in itself conservative because the authors only take the Amazon and the permafrost into account but not the ice sheet feedback (on long timescales of course) nor the feedbacks from sea-ice, etc.. While this is of course not necessary for this paper, the authors could briefly state that their estimate is conservative and may be higher if additional feedbacks would be taken into account.

This is also our understanding, we discuss this point in L 460:
*Even though we find an increase in the probabilities of triggering caused by the additional carbon emissions from carbon TEs, this impact is not strong enough to trigger any tipping cascades and remains small compared to the scenario–dependence of tipping probabilities. Even under SSP2-4.5, which features the highest long-term increase in temperature of up to 0.91°C due to carbon emissions from carbon TEs, the additional probability of instantaneous triggering caused by this temperature increase is only 3 pp on average. Despite our finding that the impact from carbon TEs alone is too small to trigger tipping cascades, tipping cascades might still emerge as major physical interactions between TEs aside from carbon emissions are not accounted for in this study (Wunderling et al., 2021).*

2) L 64: "Since the northern expansion and southern dieback … balance out". I am good with leaving this out, but it is uncertain whether it balances out in the end and probably depends on the speed of anthropogenic climate change (I assume we are faster than northern growth).

We agree that there is a lot of uncertainty whether the two will balance out or not, we rely on the estimate of Armstrong et al. (2023) for making this statement. Nevertheless, even if the two won't cancel out, their net emissions will likely be small compared to what could be expected from permafrost degradation.

Some additional and very new literature that may be worthwhile considering in the current version of the manuscript:
3) Around L 70: Here, a new review on (no) permafrost tipping points could be cited: Nitzbon, J., Schneider von Deimling, T., Aliyeva, M., Chadburn, S.E., Grosse, G., Laboor, S., Lee, H., Lohmann, G., Steinert, N.J., Stuenzi, S.M. and Werner, M., 2024. No respite from permafrost-thaw impacts in the absence of a global tipping point. Nature Climate Change, pp.1-13.
4) L87: The paper backing up the science panel for the amazon reference (2021) is here: Flores, B.M., Montoya, E., Sakschewski, B., Nascimento, N., Staal, A., Betts, R.A., Levis, C., Lapola, D.M., Esquível-Muelbert, A., Jakovac, C. and Nobre, C.A., 2024. Critical transitions in the Amazon forest system. Nature, 626(7999), pp.555-564.
5) L 97-99: I agree with the authors that timescales of tipping elements are still very difficult to assess; somewhat similarly to predicting the tipping times in the first place: Ben-Yami, M.,

Morr, A., Bathiany, S. and Boers, N., 2024. Uncertainties too large to predict tipping times of major Earth system components from historical data. Science Advances, 10(31), p.eadl4841.
6) L 455: Add the new GRIS tipping point paper by Bochow et al, 2023, Nature (basically a follow up of the earlier Robinson et al., 2012 paper): Bochow, N., Poltronieri, A., Robinson, A., Montoya, M., Rypdal, M. and Boers, N., 2023. Overshooting the critical threshold for the Greenland ice sheet. Nature, 622(7983), pp.528-536.
7) A study that couples FaIR with tipping probabilities under overhoots: Möller, T., Högner, A.E., Schleussner, C.F., Bien, S., Kitzmann, N.H., Lamboll, R.D., Rogelj, J., Donges, J.F., Rockström, J. and Wunderling, N., 2024. Achieving net zero greenhouse gas emissions critical to limit climate tipping risks. Nature Communications, 15(1), p.6192.

Thanks a lot for bringing up this new literature, it's highly appreciated! We cited the papers where we thought it was fitting, mostly following your suggestions:
Nitzbon et al., 2024 in L73
Flores et al., 2024 in L87
Bochow et al., 2024 in L452

**Response to Reviewer 2**

Thank you for taking the time to review our paper three times now. Your comments were very helpful for making this a better paper.